# Never Revisit: Continuous Exploration in Multi-Agent Reinforcement Learning

## Abstract

Recently, intrinsic motivations are wildly used for exploration in multi-agent reinforcement learning. However, we discover that coming with intrinsic rewards is the issue of revisitation – the relative values of intrinsic rewards, estimated based on neural networks, fluctuate during learning, causing failures of rediscovering promising areas after detachment of exploration. Consequently, agents risk exploring some sub-spaces repeatedly and being stacked nearing the fixed initial point. In this paper, we formally define the concept of revisitation, based on which we propose an observation-distribution matching approach to detect the appearance of revisitation. According to each detected revisitaion, we dynamically augment branches for agents' local Q-networks and the mixing network to achieve sufficient representational capacity. Furthermore, we use historical joint observations to adjust intrinsic rewards to reduce the probability of and penalize the occurrence of revisitation. By virtue of these advances, our method achieves superior performance on three challenging Google Research Football (GRF) scenarios and three StarCraft II micromanagement (SMAC) maps with sparse rewards.

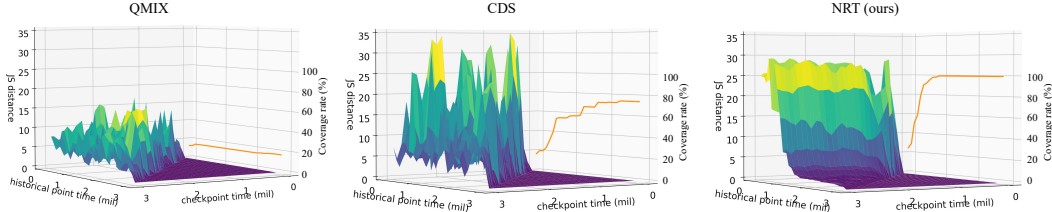

Figure 1: The issue of revisitation. During learning, we will store the joint observation distribution induced by the joint policy $\pi$ every 100k time steps, named historical point time. Meanwhile, for every 100k time step, we will calculate the JS distance between the current distribution and all historical points, named checkpoint time. Experiments are carried out on a $6 \times 12$ maze task introduced in Sec. 4. **Left**: Basic exploration scheme (QMIX (Rashid et al., 2018) with $\epsilon$-greedy exploration) achieves a coverage rate of the joint observation space less than $20\%$. **Middle**: Adding CDS (Li et al., 2021) intrinsic incentives improves the coverage rate to $80\%$. However, the fluctuating JS distances indicate that agents are periodically revisiting some sub-spaces. Consequently, most samples are wasted on revisitation. **Right**: Revisitation is avoided based on our method, and the JS distances are large and stable before converging to an optimal strategy, indicating continuous exploration achieved by our approach.

## 1 Introduction

Multi-agent cooperation is ubiquitous in real-world problems, such as sensor networks (Zhang & Lesser, 2013) and traffic light control (Zhang et al., 2019). To introduce intelligence into multi-agent systems and achieve sophisticated cooperative behavior, multi-agent reinforcement learning (MARL) has been gaining increasing interest in recent years. Advanced MARL methods have largely pushed forward the performance of machine learning algorithms on tasks such as StarCraft II micromanagement (Rashid et al., 2018; Wang et al., 2021b), Hanabi (Bard et al., 2020; Foerster et al., 2019), and robotic control (Kurin et al., 2020).

Despite these achievements, a problem persists and prevents MARL from extending successfully to more complex problems. The action-observation space grows exponentially with the number of agents, and the efficiency of exploration strategies in such search spaces largely limits the learning efficiency of MARL algorithms. Basic exploration (Wang et al., 2020b) schemes, like $\epsilon$-greedy, adopted by many previous works (Wang et al., 2021a; Yu et al., 2021; de Witt et al., 2020) appear to struggle even in tasks with a moderate number of agents. For example, in a $6 \times 12$ maze game with two agents (Fig. 4 (b)), QMIX (Rashid et al., 2018) using independent search can only cover less than $20\%$ of the joint observation space (Fig. 1 left) and struggles to find any rewards.

Various intrinsic motivations are introduced into MARL algorithms to enhance their exploration ability by encouraging interaction among agents (Wang et al., 2020b), maximizing a measurements of behavioral randomness (Houthooft et al., 2016; Mahajan et al., 2019; Gupta et al., 2021), and spurring individuality (Jiang & Lu, 2021) and diversity (Li et al., 2021). These methods significantly enlarge the sub-space that can be explored. For example, the coverage rate increases from $\sim 20\%$ to $\sim 80\%$ using the CDS (Li et al., 2021) diversity-encouragement incentives (Fig. 1 middle). However, in this paper, we find that coming with the augmented exploration is a ***revisitation*** problem that severely hurts the expected exploration ability and prevents learning efficiently.

Revisitation refers to the situation where, with an enlarged exploration space, agents forget the areas they have visited before after the detachment (Ecoffet et al., 2019) of exploration, causing them to return and re-explore. In this way, agents are stacked nearing the fixed initial point and can not explore continuously. Meanwhile, revisitation in some sub-spaces can repeatedly happen, making this issue more detrimental. For example, in Fig. 1 (middle), we present the empirical joint observation distribution induced by CDS (Li et al., 2021) policies at different training time steps. It can be observed that similar distributions occur periodically during learning, and the algorithm wastes most training samples on unvalued, revisited experiences.

In this paper, we give a formal definition of revisitation in multi-agent settings, according to which we propose a framework for solving this issue. To provide sufficient policy representation capacity, we add branches to agents' local Q-networks and the mixing network when revisitation occurs to achieve a dynamically growing neural network structure. Besides that, we use historical samples to calculate joint observation novelty for adjusting intrinsic rewards. Meanwhile, for each revisitation, we further introduce a KL divergence between its historical point's recorded joint observation distribution and the current one to penalize it to happen again.

Based on the above merits for detecting and preventing revisitation, our approach NRT (**N**ever **R**evisi**t**) achieves nearly $100\%$ coverage rate on the maze task (Fig. 1 right) with continuous exploration (Fig. 4 in Sec. 4). Furthermore, we benchmark our approach on both Google Research Football (GRF, Kurach et al. (2020)) and SMAC (Samvelyan et al., 2019) in the sparse reward setting, and find that NRT significantly pushes forward state of the art. Ablation studies show that adding branches is the most important component in revisitation avoidance and performance improvement for challenging tasks. At the same time, the proposed intrinsic reward adjustment modules are critical in achieving continuous exploration.

## 2 BACKGROUND

In this paper, we formulate multi-agent coordination tasks as Dec-POMDP Oliehoek et al. (2016), which can be defined as a tuple $\mathcal{G} = \langle I, S, A, P, R, O, n, \gamma \rangle$, where $I$ is the set of agents, $S$ is the state space, $A$ is the action space, $P$ is the transition function, $R$ is the reward function, $O$ is the observation space, $n$ is the number of agents, and $\gamma \in [0, 1)$ is the discount factor. During sampling, each agent $i \in I$ observes its unique information $o_i$ and selects an action $a_i \in A$ independently. According to the joint action $\boldsymbol{a}$ and environment transition function $P(s'|s, \boldsymbol{a})$, the environment transfers to a new state $s'$ and provides a reward that is shared across all agents.

### 2.1 CENTRALIZED TRAINING WITH DECENTRALIZED EXECUTION

Our approach follows a recently advanced multi-agent optimization framework of centralized training with decentralized execution (CTDE) Lowe et al. (2017); Foerster et al. (2018); Sunehag et al. (2018); Rashid et al. (2018). In this framework, each agent executes only based on its local action-observation history. This distributed decision-making tackles the exponentially growing joint action space. For

stable optimization, agents are trained in a centralized manner with access to global information to achieve reasonable credit assignment across agents. The individual-global-max (IGM) principle Son et al. (2019) further guarantees the consistency between the local and global Q-function based on value factorization. With the above advantages, some algorithms Son et al. (2020); Wang et al. (2021a); Ma et al. (2021) have achieved remarkable progress in challenging cooperation tasks.

## 2.2 CDS

Although the CTDE framework has been proven to accelerate training with value factorization, it still has to face a giant search space. Diversity in exploration is required for sophisticated cooperation. Li et al. (2021) introduced CDS, which achieves diversity when necessary via shared experiences. Some of the most salient novelties are: (1) Introduce an intrinsic reward function as shown in Eq. 1 based on $I^\pi(\tau_T; id)$ to encourage specialty of individual trajectories. (2) Provide an independent Q-network for each agent to estimate local Q-value after combining with one shared Q-network. (3) Add an L1 regularization on independent Q-values to encourage sharing crucial knowledge while keeping enough policy representation capacity of necessary diversity.

$$r_{\text{CDS}}^I = \beta E_{id} \left[ \beta_2 D_{\text{KL}} \left( \text{SoftMax} \left( \beta_1 Q \left( \cdot \mid \tau_t, id \right) \right) | p \left( \cdot \mid \tau_t \right) \right) \\ + \beta_1 \log q_\phi \left( o_{t+1} \mid \tau_t, a_t, id \right) - \log p \left( o_{t+1} \mid \tau_t, a_t \right) \right], \tag{1}$$

where id is the identity of each agent, $\beta$, $\beta_1$ and $\beta_2$ are hyper-parameters introduced in CDS, and $\tau_T = (o_0, a_0, o_1, \ldots, o_T)$ is the variable of one agent's trajectory. In this paper, we use $r_{\text{CDS}}^I$ as an example to expose and solve one crucial potential problem of existing intrinsic motivation methods, revisitation as shown in Fig. 1 (middle).

## 3 METHOD

### 3.1 REVISITATION DETECTION

We first define the concept of **_revisitation_** in multi-agent settings. Our definition is based on the empirical joint observation distribution induced by the joint policy $\boldsymbol{\pi}$: $\rho_{\boldsymbol{\pi}}(\boldsymbol{o}) = \mathbb{E}_{\boldsymbol{\pi}}[\frac{1}{T} \sum_{t=0}^{T} \mathbf{1}_{o_t=o}]$. Intuitively, when $\rho_{\boldsymbol{\pi}}(\boldsymbol{o})$ changes significantly, the agent team starts to explore a new sub-space. Still, if, after a significant change, $\rho_{\boldsymbol{\pi}}(\boldsymbol{o})$ degenerates to a distribution that has occurred before, revisitation happens. Formally, we have the following definition:

**Definition 1.** *A $\delta$-revisitation occurs at time step $t$ if there exists a time step $t'' < t$ so that*

1. $\max_{t' \in (t'', t)} D_{\text{JS}} \left[ \rho_{\boldsymbol{\pi}_{t''}}(\boldsymbol{o}) \| \rho_{\boldsymbol{\pi}_{t'}}(\boldsymbol{o}) \right] > \delta$;

2. $D_{\text{JS}} \left[ \rho_{\boldsymbol{\pi}_{t''}}(\boldsymbol{o}) \| \rho_{\boldsymbol{\pi}_t}(\boldsymbol{o}) \right] < \delta$.

Here, $D_{\text{JS}}$ is the Jensen-Shannon distance between two distributions. The first condition describes the ability of many multi-agent exploration algorithms. As the learning progresses, agents are encouraged to explore new sub-spaces, leading to a shift in the empirical joint observation distribution. The second condition depicts the occurrence of revisitation. After exploring a new sub-space for a while, the agents may forget previously learned knowledge, possibly due to limited network representative capacity, and start to re-explore previously visited sub-spaces.

Under discrete observation space, we can estimate $\rho_{\boldsymbol{\pi}}(\boldsymbol{o})$ with the Monte Carlo (MC) method. However, in continuous observation space, it is intractable to use MC. In practice, we estimate the joint observation distribution $\rho_{\boldsymbol{\pi}}(\boldsymbol{o})$ using an a variational auto-encoder parameterized by $\xi$, by minimizing the evidence lower bound:

$$\mathcal{L}_\rho(\xi) = \mathbb{E}_{q(z|\boldsymbol{o};\xi)}[\log p(\boldsymbol{o}, z; \xi) - \log q(z \mid \boldsymbol{o}; \xi)], \tag{2}$$

where $z$ is a latent variable to reconstruct $\boldsymbol{o}$, and $p_\xi(\boldsymbol{o}) = \int p_\xi(\boldsymbol{o} \mid z) p(z) \mathrm{d}z \approx \frac{1}{L} \sum_{i=1}^{L} p_\xi(\boldsymbol{o} \mid z_i)$ for $z_i \sim p(z)$, where $p(z)$ is a prior of the latent variable vector, which is modeled by a standard Gaussian distribution $\mathcal{N}(z \mid \mathbf{0}, \mathbf{I})$ and $L$ is fixed as 32 in this paper.

During training, for every 100k time step, we store the current joint observation distribution estimator $p_{\xi_t}$ into a buffer, and check for each previous version $p_{\xi_{t''}}$ whether they satisfy the two conditions

listed in Definition 1. If the conditions are satisfied, we call $(t'', t', t)$ a **revisitation tuple** and store $(\rho_{\boldsymbol{\pi}_{t''}}(\boldsymbol{o}), \rho_{\boldsymbol{\pi}_{t'}}(\boldsymbol{o}), \rho_{\boldsymbol{\pi}_t}(\boldsymbol{o}))$ with $(\rho_{\boldsymbol{\pi}_{t''}}(o_i), \rho_{\boldsymbol{\pi}_{t'}}(o_i), \rho_{\boldsymbol{\pi}_t}(o_i))$ for each agent $i \in I$, where $\rho_{\boldsymbol{\pi}}(o_i) = \mathbb{E}_{\boldsymbol{\pi}} \left[ \frac{1}{T} \sum_{t=0}^{T} \mathbf{1}_{o_{i,t}=o_i} \right]$ is estimated along with $\rho_{\boldsymbol{\pi}}(\boldsymbol{o})$.

## 3.2 AUGMENTATION OF THE TRAINING FRAMEWORK

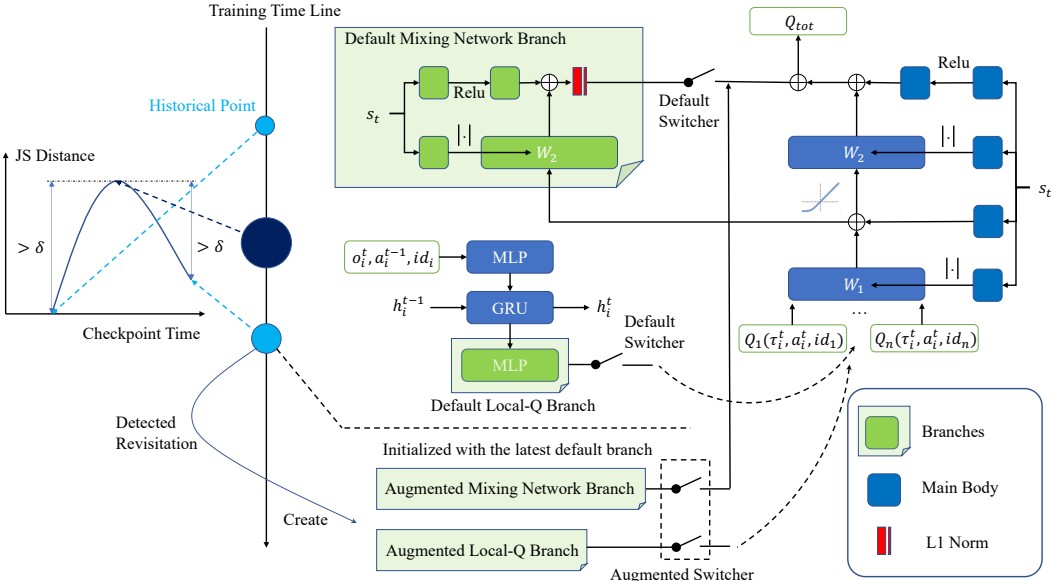

Figure 2: Dynamic augmentation of NRT (**N**ever **R**evisi**t**)'s neural network structure (using one detected revisitation as an example).

To achieve sufficient policy representation capacity for handling revisitation while benefiting from parameter sharing, we dynamically augment partial neural networks (branches) as shown in Fig. 2. In this paper, we follow the CTDE framework as discussed in Sec. 2.1 with a mixing network as QMIX (Rashid et al., 2018), whose weights are produced by separate hyper networks with an input of state and are followed by an absolute activation function to ensure the IGM principle. Starting from the beginning of training, we provide a default mixing network branch (a part of the mixing network), a default local-Q branch (the last layer of local Q-networks), and the main body of our neural networks. The main body is painted blue, while branches are painted green. After one detection of revisitation, we will create a related augmented mixing network branch and a related augmented local-Q branch with the same structure as the default ones to increase representational capacity. Their parameters are initialized according to the latest default branches.

Along with our dynamic augmentation structure is a question about how to switch different branches. Formally, for $b$ th augmented branch, we define the associated joint observation set $\Omega_b$ based on the definition of related revisitation tuple $(t_b'', t_b', t_b)$ introduced in Sec. 3.1.

$$\Omega_b = \{\boldsymbol{o} \big| |\log \rho_{\boldsymbol{\pi}_{t_b''}}(\boldsymbol{o}) - \log \rho_{\boldsymbol{\pi}_{t_b'}}(\boldsymbol{o})\}| > \delta\} \cap \{\boldsymbol{o} \big| |\log \rho_{\boldsymbol{\pi}_{t_b''}}(\boldsymbol{o}) - \log \rho_{\boldsymbol{\pi}_{t_b}}(\boldsymbol{o})\}| < \delta\}. \quad (3)$$

However, during decentralized execution, one agent can only obtain its observation. Thus, we further introduce the marginal set $\Omega_b^i$ for each agent $i \in I$ as shown in Eq. 4. If agent $i$'s local observation $o_i \in \Omega_b^i$, the related $b$ th augmented local-Q branch will be activated for both sampling and training. If none of augmented branches is activated or created, agents will choose the default branch.

$$\Omega_b^i = \{o_i \big| |\log \rho_{\boldsymbol{\pi}_{t_b''}}(o_i) - \log \rho_{\boldsymbol{\pi}_{t_b'}}(o_i)\}| > \frac{\delta}{n}\} \cap \{o_i \big| |\log \rho_{\boldsymbol{\pi}_{t_b''}}(o_i) - \log \rho_{\boldsymbol{\pi}_{t_b}}(o_i)\}| < \frac{\delta}{n}\}. \quad (4)$$

There are two questions coming with Eq. 4, (1) how to switch augmented local-Q branches when $o_i$ belongs to several sets simultaneously, and (2) how to switch augmented mixing network branches when agents switch various local-Q branches. As for question (1), we will check according to the created order and activate the earliest branch. As for question (2), each augmented mixing network branch will be activated only when more than half of agents choose the corresponding augmented local-Q branch. Our neural networks will switch to the default branch if none of augmented branches is activated or created. Because the whole trajectory is spliced from different branches, we expect credit assignments across agents (achieved by the mixing network) to not differ too much for stability. Thus we add an L1 regularization to mixing network branches to encourage them to work only when necessary. More details and the pseudo-code of NRT's sampling process are discussed in Sec. B.1

### 3.3 AUXILIARY SUPPORTING INCENTIVES

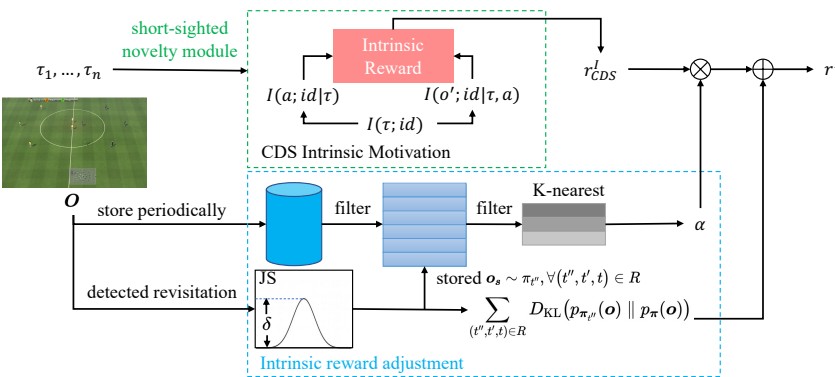

Figure 3: NRT (**N**ever **R**evisi**t**)'s intrinsic reward adjustment modules.

A possible shortcoming of the methods introduced above is that when revisitation occurs frequently, there may be too many branches. Thus, we design adjustment modules for intrinsic motivations to attempt to (1) prevent the appearance of revisitation in advance, and (2) penalize the occurrence of revisitation.

To prevent revisitation in advance, we adjust intrinsic rewards with historical joint observations. In this paper, we periodically store joint observation sequences (every ten episodes) in an independent buffer without deletion. During training, for time efficiency, we form a compact sub-sampling buffer with 10 trajectories from the whole learning period. As inspired by Badia et al. (2020), based on joint observation novelty estimated by K-nearest approach, we calculate the weight of intrinsic rewards: $\alpha = \frac{1}{\sqrt{\sum_{o_{\mathcal{N}} \in \mathcal{N}_k(o)} K(o, o_{\mathcal{N}}) + \epsilon}}$, where $\mathcal{N}_k(o)$ is the $k$ nearest neighbors of $o$ in the compact buffer, and $\epsilon$ is a small constant (0.001) for numeric stability. The kernel function $K(\cdot, \cdot)$ is defined as $K(x, y) = \frac{\epsilon}{d^2(x,y)+\epsilon}$, where $d$ is the Euclidean distance.

For penalizing revisitation to forbid it to occur again, we additionally add 5 trajectories from each revisitation to the compact sub-sampling buffer discussed above to refresh the estimated joint observation novelty. Furthermore, we introduce $r^p = \sum_{(t'',t',t) \in R} D_{\text{KL}}\left(p_{\boldsymbol{\pi}_{t''}}(\boldsymbol{o}) \| p_{\boldsymbol{\pi}}(\boldsymbol{o})\right)$, where $p_{\boldsymbol{\pi}}$ is the current joint policy. Overall, the intrinsic reward used in our paper is

$$r^I = r^I_{\text{CDS}} \cdot \texttt{clip}(1 + \bar{\alpha}, L_1, L_2) + \beta_p r^p, \tag{5}$$

where $\bar{\alpha} = \frac{\alpha - \mu_e}{\sigma_e}$ is the running average of $\alpha$, $\texttt{clip}$ means we clip the running average so that it is in the range $[L_1, L_2]$, $r^I_{\text{CDS}}$ is the intrinsic reward introduced by CDS (Li et al., 2021) as shown in Eq. 1, and $\beta_p$ is a hyper-parameter tuning the weight of $r^p$, which is fixed as 0.001 in our experiments.

## 4 DIDACTIC EXAMPLE

In this section, we present a maze task (shown in Fig. 5 (b)) to demonstrate the existence of revisitation, its influence on exploration, and how our approach improves learning performance by

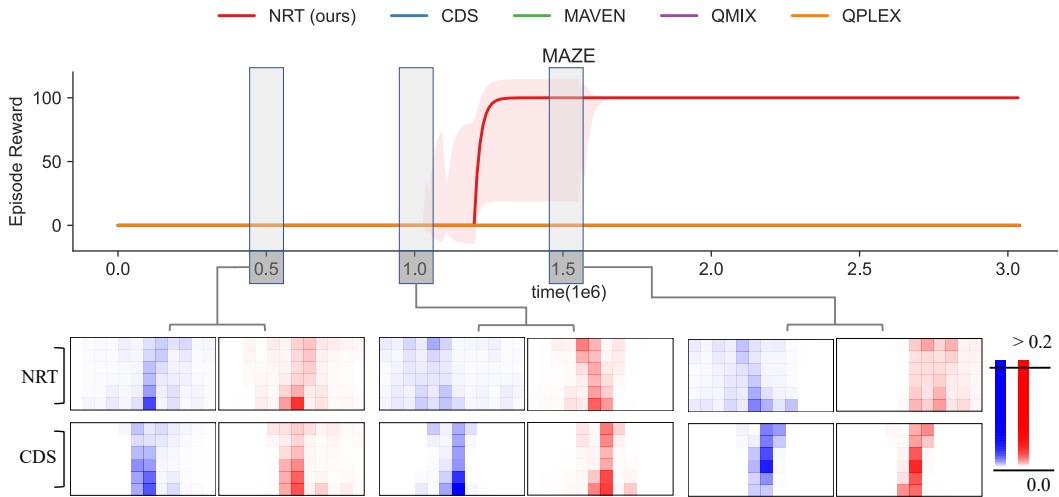

Figure 4: Comparison against baselines with visitation heat maps.

avoiding revisitation. In this task, two agents are initialized at the blue point and are expected to simultaneously reach two red points located in different corners of a $6 \times 12$ maze within 50 steps. Each agent can observe its own and teammate's coordinates. The action set of both agents includes one-step movement in four directions and an idle action for doing nothing. Only when agents reach two red points simultaneously, the episode will terminate immediately with a team reward of 100. For any other cases, reward is set to 0.

Fig. 4 shows the comparison of our approach against baseline algorithms, including state-of-the-art value function factorization learning methods (QMIX (Rashid et al., 2018) and QPLEX (Wang et al., 2021a)) and multi-agent exploration algorithms (CDS (Li et al., 2021) and MAVEN (Mahajan et al., 2019)). Experimental results demonstrate the advantage of our approach in exploration, as NRT is the only algorithm that obtains the episode reward of 100. We further provide visitation heat maps of NRT and CDS within three periods ($0 \sim 0.5M$, $0.5M \sim 1.0M$, and $1.0M \sim 1.5M$). NRT continuously encourage agents to explore further, while agents trained by CDS are stacked nearing the fixed initial points and waste most training samples on revisited experiences after the detachment of previous exploration.

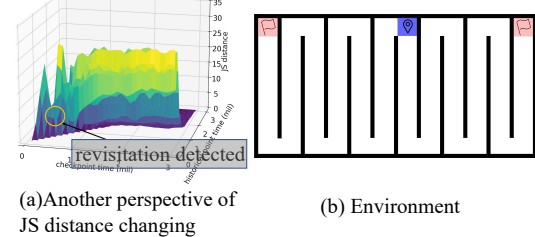

(a)Another perspective of JS distance changing

(b) Environment

Figure 5: Structure of the maze and continuous exploration achieved by our approach.

In Fig. 5 (a), we further show another perspective of the 3D plot of JS distance plot to demonstrate continuous exploration achieved by our approach after detecting revisitation. The circle in Fig. 5 (a) highlights a time step when a revisitation is detected. At this time, our approach (1) adds a new branch as described in Sec. 3.2; (2) ensures that the joint observations where the revisitation happens are sampled when calculating the intrinsic reward; and (3) adds punishment of revisitation. In this way, our approach effectively prevents the reappearance of revisitation, leading to continuous exploration.

## 5 EXPERIMENT

In Sec. 4, we introduce a toy maze environment to illustrate our approach's efficiency of exploration while showing the significant harm of repeat revisitation learned by current advanced multi-agent algorithms. In this section, we will compare our approach against baselines on Google Research Football (GRF) Kurach et al. (2020) and SMAC (Samvelyan et al., 2019) in the sparse reward setting, which are currently the most challenging benchmarks for testing cooperation between agents. We show the median and variance of every evaluation's average performance of our approach, baselines, and ablations tested with three random seeds (seed=0,1,2).

## 5.1 GRF

The GRF benchmark provides numerous scenarios varying in agent numbers and difficulty. In this paper, we use three representative scenarios to demonstrate the significance of considering revisitation. Agents' initial locations for each scenario are shown in Appendix A. Following the convention of Kurach et al. (2020) and Li et al. (2021), the state and each agent's observation are based on the official simple 115 representation vector, while observations are established according to relative positions. Each agent has a discrete action space of 19 dimensions, including moving and several basic actions, such as passing and shooting. Environmental rewards are only provided at the end of one episode. Agents will get $+100$ if they score, else get $-1$.

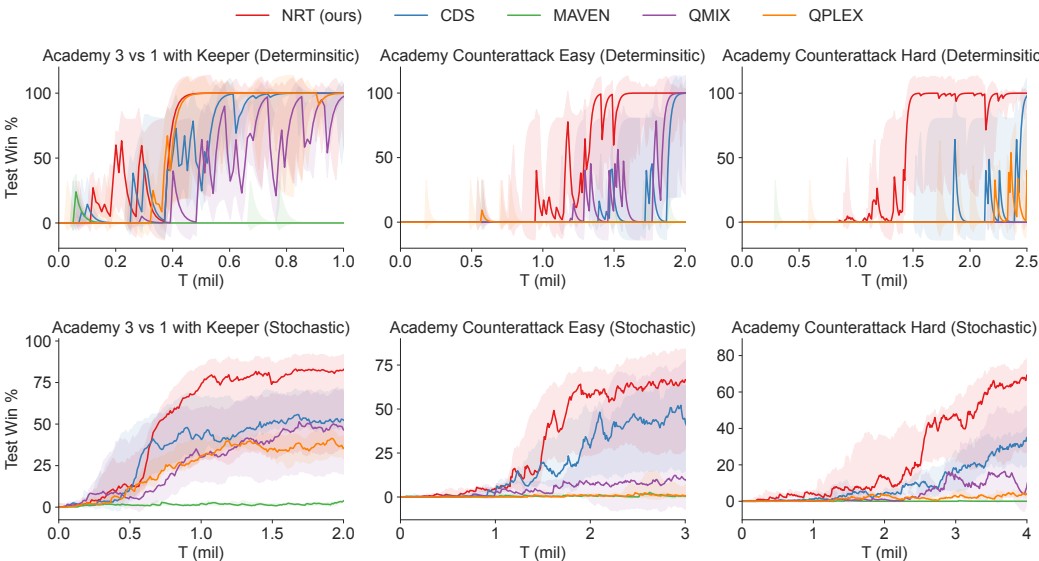

Figure 6: Comparison of our approach against baseline algorithms on GRF with deterministic environment seeds or stochastic environment seeds.

Fig. 6 demonstrates the comparison between our approach with baseline algorithms in the deterministic and stochastic settings. Here we first analyze experiment results in the deterministic setting. In `Academy_3_vs_1_with_Keeper`, QPLEX can achieve similar learning efficiency with our approach, while CDS and QMIX can also achieve a 100% winning rate. However, in more complex scenarios, `Academy_Counterattack_Hard` in particular, the outperformance of our approach is obvious, showing the advantages of considering revisitation. In `Academy_3_vs_1_with_Keeper` and `Academy_Counterattack_Easy`, CDS performs similar with QMIX. With our additional supporting incentives, NRT achieves more efficient exploration.

Like the deterministic counterpart, our approach maintains its outperformance in the stochastic setting, as shown in Fig. 6. Because opposing players' behavior might differ across various random environment seeds, the average winning rates during evaluation are nearly impossible to be 100%. In this case, our approach still shows clear superiority in all three scenarios. In challenging scenarios, such as `Academy_Counterattack_Easy`, CDS outperforms QMIX. Following the guidance of our intrinsic reward adjustment module, the performance is further raised in all three scenarios.

**Ablation Study** To understand the contribution of each component in the proposed NRT framework, we conduct three ablations: (1) ablate augmented branches, (2) ablate auxiliary dynamic weight of intrinsic motivations, and (3) ablate auxiliary KL punishment after the detection of revisitation. We respectively name them NRT-ab-new-branch, NRT-ab-alpha, and NRT-ab-KL. To further understand the necessity of detecting revisitation, we conduct two ablations based on vanilla CDS: (1) store joint observation distributions every 100k time steps and add the KL divergence between each of them and the current one as intrinsic rewards during training, (2) estimate the distribution of all sampled joint observations and add the KL divergence between it and the current one as intrinsic rewards during training. We respectively name them CDS-KL-Each and CDS-KL-All.

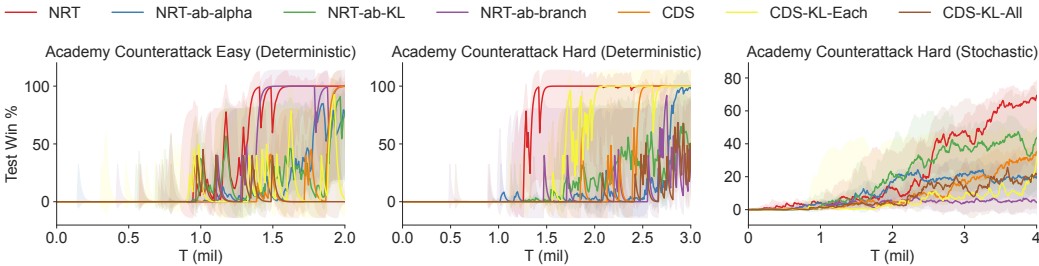

Figure 7: Ablation studies on three GRF tasks.

We compare NRT with its ablations on three representative tasks: `Academy_Counterattack_Easy` under the deterministic setting and `Academy_Counterattack_Hard` under both settings. From Fig. 7 left to Fig. 7 right, the difficulty of the environment increases, accompanied by a sharp decline in the performance of NRT-ab-new-branch. This phenomenon reveals the importance of constructing dynamic augmentation of neural networks according to the degree of exploration in complex environments. The performance of NRT-ab-alpha and NRT-ab-KL in three tasks illustrates that the combined effect of our auxiliary supporting incentives makes our approach outperform CDS. CDS-KL-Each performs well in `Academy_Counterattack_Hard` under the deterministic setting but fails on the other two scenarios, while CDS-KL-All performs similarly to vanilla QMIX. This phenomenon demonstrates the necessity of dynamically checking the exploration situation and making targeted adjustments when revisitation occurs.

## 5.2 SMAC

In this section, we test our approach on SMAC, consisting of various maps with different agent numbers and various agent types. Previous work has achieved remarkable performance in many challenging tasks. However, their success heavily relays on dense rewards provided by the environment, which includes changes in the blood volume of our agents and opposing agents. In this paper, we consider SMAC tasks in the sparse reward setting. Agents will only receive rewards at the end of one episode. Agents will get +100 if they win, else get -1, which maintains the same as that in GRF.

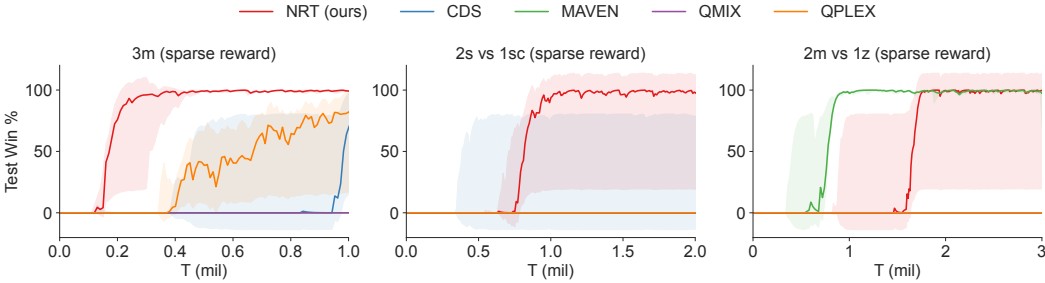

Figure 8: Comparison of our approach against baseline algorithms on SMAC in the sparse reward setting.

Fig. 8 demonstrates the comparison between our approach with baseline algorithms in three easy SMAC maps with sparse reward. QMIX achieves good performance with dense rewards (Rashid et al., 2018). However, vanilla QMIX can not learn any strategies to win in all three easy maps based on sparse rewards as shown in Fig. 8. Using QMIX as the mixing network, CDS can learn valuable strategies in 3m with its intrinsic motivations as discussed in Sec. 2.2. However, CDS still learns nothing to win in `2m_vs_1z`. Our approach, adjusting intrinsic rewards of CDS based on the detection of revisitation, outperforms CDS in all three maps. The comparison between NRT and CDS supports our above discussion about one potential drawback of current intrinsic motivation methods, revisitation, which seriously influences exploration efficiency. QPLEX achieves good performance in 3m with its duplex dueling network architecture, but still learns nothing to win in other two maps.

MAVEN achieves remarkable performance in 2m_vs_1z, but performs similar with QMIX in other two maps. Compared with QPLEX and MAVEN, NRT can adapt to all three maps, showing its exploration efficiency.

## 6 RELATED WORK

**Exploration Bonuses.** Exploration is one key factor that determines the efficiency of deep reinforcement learning (Houthooft et al., 2016; Trott et al., 2019; Raileanu & Rocktäschel, 2020; Zintgraf et al., 2021; Chen et al., 2021). Starting from the classic $\epsilon$-greedy action selector, numerous advanced algorithms are proposed to enhance the efficiency of exploration. Count-based exploration methods directly use visit counts to guide an agent's behavior towards reducing uncertainty (Strehl & Littman, 2008; Bellemare et al., 2016; Martin et al., 2017; Ostrovski et al., 2017; Tang et al., 2017; Machado et al., 2020). A representative method is Random Network Distillation (RND), using prediction errors to estimate count-based rewards (Burda et al., 2018b; Osband et al., 2018; 2019; Ciosek et al., 2019). The prediction errors are also used in Pathak et al. (2017); Burda et al. (2018a); Pathak et al. (2019); Dean et al. (2020) to achieve self-organized intrinsic motivations.

**Exploration in the Multi-Agent Setting.** Exploration is more crucial in the multi-agent setting, because the search space grows exponentially with the number of agents. Besides exploring the environment, we also need to consider diversity across agents to achieve sophisticated coordination. In the multi-agent setting, agents' exploration can be jointly encouraged (Mahajan et al., 2019; Gupta et al., 2021; Zheng et al., 2021; Liu et al., 2021), encouraged in pairs (Wang et al., 2020b; Ndousse et al., 2021), or independently encouraged for being special from the whole group (Jiang & Lu, 2021; Li et al., 2021). Exploration can also be encouraged through the factorization of roles (Wang et al., 2020a; 2021b). In this paper, we choose CDS (Li et al., 2021) as the intrinsic motivation for exploration because it is the current state-of-the-art algorithm in many challenging benchmarks.

**Forgetting.** Revisitation discussed in our paper also relates to the forgetting of previous policies, which is wildly discussed in lifelong learning with the multi-task setting (Rebuffi et al., 2017; Li & Hoiem, 2017; Rolnick et al., 2019; Von Oswald et al., 2019). Some advanced algorithms have studied this topic by extending the supervised regularization (Kirkpatrick et al., 2017) or replay (Isele & Cosgun, 2018; Yan et al., 2022) paradigms, exacerbating the stability-plasticity tension. Others, on the other hand, have proposed multi-stage processes in which the agent first applies existing knowledge to the current task and then incorporates newly acquired knowledge into a shared repository (Schwarz et al., 2018; Mendez et al., 2020). However, forgetting is rarely discussed in single-task and multi-agent reinforcement learning, which also has the possibility of forgetting during exploration. The key question is how to detect and forbid it automatically without any information about the stages agents are in during learning.

In this paper, we attempt to answer this question to enhance exploration efficiency. Ecoffet et al. (2019) and Ecoffet et al. (2021) intuitively show a hypothetical example of revisitation caused by detachment in intrinsic motivations. They solve it by storing all visited states and related policies to access. Before sampling, they will first check which state to return from the whole state space, which is challenging to execute in a large continuous search space, especially in multi-agent settings. In this paper, we empirically show policies' repeated revisitation in the multi-agent setting and solve it based on much fewer memories. Our approach maintains active detection and prevention of revisitation. We hope this paper can stimulate a rethinking of intrinsic motivations for more effective exploration in single-task multi-agent learning.

## 7 CLOSING REMARKS

In this paper, we investigate the intrinsic motivation methods for multi-agent exploration, and find the revisitation issue that prevents these methods from achieving efficient exploration in complex tasks based on sparse rewards. For future work, we plan to study the interplay between the number of augmented branches and the scale of the proposed intrinsic rewards. We hope our work can encourage more studying on the limitation of current exploration methods based on intrinsic motivations to push forward the boundary of artificial intelligence.

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

## A    EXPERIMENT SETTING

The visualization of the initial position of each agent is shown in Fig. 9. The difficulty of scenarios increases from `Academy_3_vs_1_with_Keeper` to `Academy_Counterattack_Hard` based on a noticeable difference between the number of agents and the initial distance of each agent to the goal and opposing players.

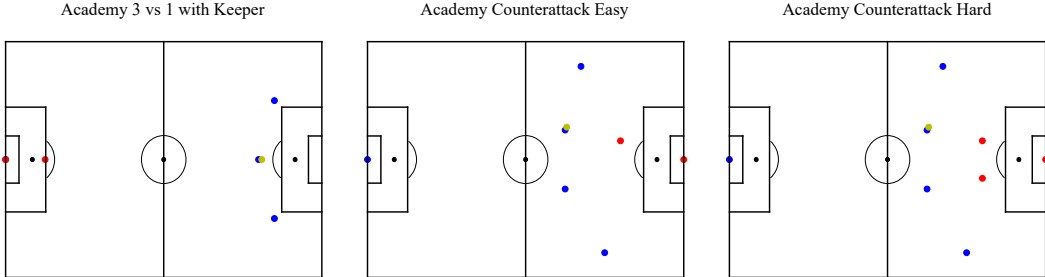

Figure 9: Visualization of the initial position of each agent in three GRF scenarios considered in our paper, where blue points represent our controlled team, red points represent the opposing team, and yellow point represents the ball.

## B    ARCHITECTURE AND HYPER-PARAMETERS

In this paper, we use simple network structures for the local Q-networks and the mixing networks as baselines. Agents use a partially shared module to represent local Q-functions as CDS (Li et al., 2021). Following the setting of CDS, agents share a trajectory encoding network made up of two layers: a fully connected layer followed by a GRU layer with a 64-dimensional hidden state for individual Q-functions. Following the trajectory encoding network, all agents share a one-layer Q network, with each agent having its own Q network with the same structure as the shared Q network. To estimate the global action values, we use QMIX-style mixing networks, which include two 32-dimensional hidden layers activated by ReLU. Hypernetworks condition on global states to generate mixing network parameters. These options apply to QMIX Rashid et al. (2018) as well.

All experiments are optimized using RMSprop with a learning rate of $5 \times 10^{-4}$, $\gamma$ of 0.99, and no momentum or weight decay. We use $\epsilon$-greedy with $\epsilon$ anneals linearly from 1.0 to 0.05 over 50K time steps and keep constant for the rest of the training while selecting actions. We will evaluate the latest policies 32 times every 10k steps to calculate the average winning rate. For our approach, all baselines and ablations, we introduce a prioritized replay buffer with the same $\alpha$ of 0.3 and use TD(0.8) while calculating target values.

The intrinsic motivation introduced in CDS includes three hyper-parameters: $\beta$, $\beta_1$, and $\beta_2$, with another hyper-parameter $\lambda$ controls the weight of L1 regularization on independent local Q-networks. In this paper, we roughly fine-tune these hyper-parameters while ensuring CDS uses the same hyper-parameters during comparison. For the maze game, we use $\{\beta, \beta_1, \beta_2 \lambda\}$ as $\{0.15, 0.5, 0.5, 0.01\}$. For all GRF scenarios, we use $\{\beta, \beta_1, \beta_2, \lambda\}$ as $\{0.05, 0.5, 1.0, 0.1\}$, which is the same as the original CDS setting in GRF environments.

In our approach, we introduce several special hyper-parameters and some basic settings to establish our framework. $\delta$ in Definition 1 is fixed as 4 in all experiments. $\eta$ in Eq. 3 is fixed to 1 in all experiments. To reduce the size of the replay buffer loading trajectories to calculate $\alpha$, we will save trajectories every ten times. The weight of L1 normalization on independent mixing networks is fixed to 0.1 in all experiments. $\{L_1, L_2, \beta_p\}$ is set to $\{0.5, 2.0, 10^{-3}\}$ for the maze game. For all GRF scenarios, $\beta_p$ is set to 0.001. $\{L_1, L_2\}$ is set to $\{0.5, 2.0\}$ for the scenario `Academy_Counterattack_Easy` based on both stochastic and deterministic environment setting. Meanwhile, $\{L_1, L_2\}$ is set to $\{1.0, 2.0\}$ for other scenarios. For all SMAC maps, $\{L_1, L_2, \beta_p\}$ is set to $\{1.0, 4.0, 10^{-3}\}$.

We use the codes provided by the authors for all baseline algorithms.

## B.1 Supplementary Interpretation of NRT's Dynamically Augmented Neural Network Structure

In this paper, at the beginning of learning, when no revisitation is detected, we maintain local Q-networks as CDS (Li et al., 2021) with a QMIX-style mixing network to estimate $Q_{tot}$.

Based on each revisitation, we separate the whole joint observation space into sub-spaces with Eq. 3. In this section, for convenience, we use sub-space to represent sub- joint observation space which is dynamically separated along with the detection of revisitaion. To provide sufficient policy representation capacity, we dynamically augment the neural network structure for each sub-space as shown in Fig. 2.

The last layers of agents' local Q-networks are independent across sub-spaces, while the previous layers are shared for benefits in learning efficiency from parameter sharing. Here we name the augmented last layer of agents' local Q-networks as the *augmented local-Q branch* activated in the sub-space of related revisitation. The original last layer of agents' local Q-networks initialized at the beginning of learning is named as *default local-Q branch* activated when no revisitation is detected or in the sub-space related to no revisitation.

For achieving sufficient representational capacity of the mixing network, we augment it along with augmentations of local Q-networks as shown in Fig. 2.

The original QMIX (Rashid et al., 2018) calculates $Q_{tot}$ with:

$$Q_{tot} = ([Q_1, ..., Q_n]W_1 + b_1)W_2^T + b_2, \tag{6}$$

where $Q_i$ is the local Q-value of agent $i \in I$, $W_1$ is a $n \times m$ matrix, $b_1$ and $W_2$ are vectors with $m$ dimensions, $b_2$ is a variable with 1 dimension. $W_1$ and $W_2$ are both with non-negative values. $W_1$, $W_2$, $b_1$, and $b_2$ are all outputs of several hypernetworks with the state as input.

However, the shared mixing network might not have enough capacity while the number of local-Q branches increasing. Thus we further expand the mixing network. Our QMIX-style mixing network is

$$\begin{aligned} Q_{tot} &= ([Q_1, ..., Q_n]W_1 + b_1)(W_2^T + W_B^T) + b_2 + b_B \\ &= (([Q_1, ..., Q_n]W_1 + b_1)W_2^T + b_2) + (([Q_1, ..., Q_n]W_1 + b_1)W_B^T + b_B) \\ &= Q_{tot}^S + Q_{tot}^I, \end{aligned} \tag{7}$$

where $W_B$ is a vector with $m$ dimensions and $b_B$ is a variable with 1 dimension. $Q_{tot}^S$ is shared among sub-spaces with shared parameters $W_1$, $b_1$, $W_2$, and $b_2$. $Q_{tot}^I$ is an individual part of each sub-space's mixing network with individual parameters $W_B$ and $b_B$. Here we name the augmented $W_B$ and $b_B$ related to each revisitation as the *augmented mixing network branch*. The original $W_B$ and $b_B$ initialized at the beginning of learning is named as *default mixing network branch* activated when no revisitation is detected or the sub-space related to no revisitation.

In its current form, the mixing network relays on $Q_{tot}^S$ or $Q_{tot}^I$ arbitrarily. On the contrary, we expect credit assignments across agents (achieved by the mixing network) to not differ too much between sub-spaces for stability. Thus we add an L1 regularization to mixing network branches to encourage them to work only when necessary.

The original QMIX (Rashid et al., 2018) train $Q_{tot}$ with

$$\mathcal{L}(\theta) = \sum_{i=1}^{b} \left[ \left( y_i^{tot} - Q_{tot}(\boldsymbol{\tau}, \mathbf{u}, s; \theta) \right)^2 \right], \tag{8}$$

where $y^{tot} = r + \gamma \max_{\mathbf{u}'} Q_{tot}(\boldsymbol{\tau}', \mathbf{u}', s'; \theta^-)$, $\theta^-$ are the parameters of a target network as in DQN, and $b$ is the batch size of transitions sampled from the replaybuffer.

In this paper, we train $Q_{tot}$ with

$$\mathcal{L}(\theta) = \sum_{i=1}^{b} \left[ \left( y_i^{tot} - Q_{tot}(\boldsymbol{\tau}, \mathbf{u}, s; \theta) \right)^2 \right] + \sum_{i=1}^{b} ||Q_{tot}^{I,-}||_1, \tag{9}$$

where (1) $y^{tot} = r^E + r^I + \gamma \max_{\mathbf{u}'} Q_{tot}(\boldsymbol{\tau}', \mathbf{u}', s'; \theta^-)$, $r^E$ is environmental rewards, and $r^I$ can be any intrinsic rewards. In this paper, we use intrinsic rewards from CDS (Li et al., 2021), which is named as $r_{\text{CDS}}^I$. (2) $Q_{tot}^{I,-} = (\texttt{stopgrad}([Q_1, ..., Q_n]W_1 + b_1))W_B^T + b_B$ is an additional regularization loss item. $\texttt{stopgrad}$ is used to stop gradient back-propagation. Thus we can only regularize parameters related to each mixing network branch.

During decentralized execution, one agent can only obtain its local observation. Thus, we introduce the marginal condition Eq. 4 based on Eq. 3 to guide each agent to select local-Q branches for sampling as shown in Algorithm 1. To maintain consistency between sampling and training, Algorithm 1 is also used to select each agent's local Q-network branches while training. To solve the question about how to switch augmented mixing network branches when agents switch various local-Q branches at one time step, each augmented mixing network branch will be activated only when more than half of agents choose the corresponding augmented local-Q branch. Otherwise, the default mixing network branch is switched.

---

**Algorithm 1** The sampling process of NRT

---

1: **while** Simulating **do**
2:    **if** no augmented local-Q branches **then**
3:       All agents activate the default local-Q branch to select actions
4:    **else**
5:       **for** agent $i \in I$ **do**
6:          **for** augmented branch id $b$ from the earliest created branch to the latest created branch **do**
7:             **if** $o_i \in \Omega_b^i$ based on Eq. 4 **then**
               Agent $i$ activates the $b$ th augmented local-Q branch to select actions and stops searching augmented branches.
8:             **end if**
9:          **end for**
10:          **if** no augmented local-Q branch is activated by agent $i$ **then**
            Agent $i$ activates the default local-Q branch to select actions.
11:          **end if**
12:       **end for**
13:    **end if**
14: **end while**
15: **Return** trajectories for training.

---

## C   Experimental Details

### C.1   JS Distances Calculated During Training in GRF

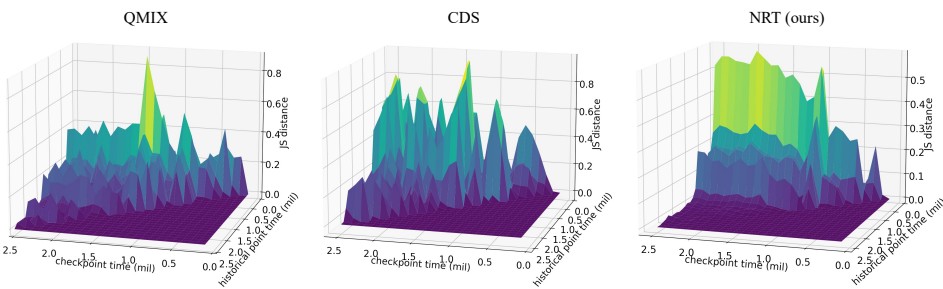

Figure 10: JS distances calculated during training in `Academy_Counterattack_Hard` under the deterministic setting

We further show JS distances calculated during training in `Academy_Counterattack_Hard` under the deterministic setting as a representative of GRF environments. The definition of the x-y axis is the same as that introduced in Sec. 4. Similar to our toy maze environment, JS distances between historical points and checkpoints periodically fluctuate under the training of QMIX and CDS, as shown in Fig. 10 left and middle. Meanwhile, under our approach's guidance of preventing revisitation, policies learned in complex environments achieve stable improvement, as shown in Fig. 10 right. Taking advantage from stable improved exploration, our approach achieves outstanding performance in challenging GRF scenarios.

## C.2   THE 2D-COLOR PLOT OF JS DISTANCES DURING TRAINING

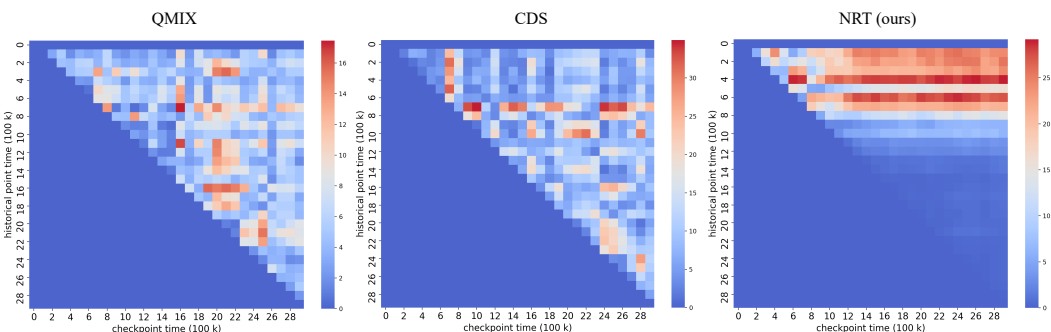

Figure 11: The 2D-color plot of Fig. 1.

We demonstrate the 2D-Color Plot of Fig. 1 as shown in Fig. 11 for readers to follow our work's motivations easier. During learning, we will store the joint observation distribution induced by the joint policy $\pi$ every 100k time steps, named historical point time (the y-axis in Fig. 11). Meanwhile, for every 100k time step, we will calculate the JS distance between the current distribution and all historical points, named checkpoint time (the x-axis in Fig. 11). Half of each picture in Fig. 11 is printed in the same color. That is because checkpoint time is always larger than historical times while calculating the JS distances. For each historical point, it is compared with checkpoints after it every 100k time steps. In Fig. 11 right, most JS distances are large and stable until coverage to the optimal strategy, after our approach detects and solves revisitation at the early period of learning.

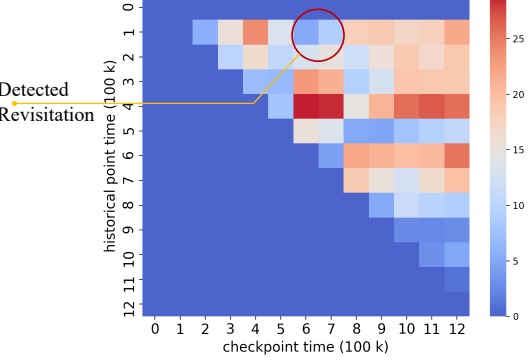

Figure 12: The 2D-color plot of JS distances between polices learned by our approach until converge.

In Fig. 12, we show and enlarge a part of Fig. 11 right to demonstrate our approach's learning progress before coverage clearly. *Through the whole learning period, revisitation means that the exploration achieved at one learning period is actually driving the joint policy to be similar to some historical policies.* For example, in Fig. 12, if we choose the joint observation distribution counted at 300k time-step as the historical point, checkpoints at 400k and 500k time-step indicate that intrinsic motivations are driving agents to explore the environment. However, if we choose the joint observation distribution counted at 100k time-step as the historical point, checkpoints at 400k and 500k time-step indicate that intrinsic motivations are driving agents to re-explore the sub-space that has been explored at 100k time-step. In this paper, we call this phenomenon revisitation and attempt to solve it with (1) augmented neural networks for sufficient policy representation capacity, (2) adjusted intrinsic motivation weight according to joint observation novelty compared with each revisitation historical point, and (3) further punishment of revisitation based on KL divergence. Based on these novelties, our approach forbids revisitation to occur again and achieves the optimal strategy in the maze exploration task by driving agents to explore away from both 100k time-step and 300k time-step, indicating reliable exploration efficiency.

As for baselines such as QMIX (explores with $\epsilon$-greedy) and CDS (explores with intrinsic motivations), revisitation occurs periodically, which heavily influences exploration efficiency and causes agents to be stacked nearing the fixed initial points as shown in Fig. 4.

## C.3  WILL THE SIZE OF REPLAY BUFFER INFLUENCE REVISITAION?

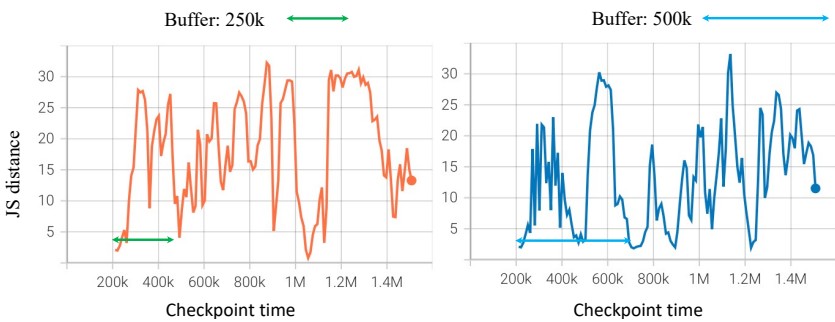

Figure 13: JS distances calculated during training with the CDS algorithm in our maze environment based on historical point at 200k time steps

Coming up with the phenomenon of revisitation based on intrinsic motivations is one natural question: Which role does the replay buffer play in revisitation? Here we still use CDS as an example of intrinsic motivations and test it in our maze environment. *Fig. 13 demonstrates that the size of replay buffer has trivial influence on revisitaion.* In Fig. 13 left, it seems that revisitation occurs according to the size of the replay buffer. However, after we multiply the replay buffer's size from 250k time steps to 500k time steps, the interval of periodical revisitation is similar to that with smaller replay buffer as shown in Fig. 13 right. Thus, in this paper, we choose to fix our replay buffer's size as baselines and adjust intrinsic motivations with augmented policy representation capacity.

## C.4  EXPERIMENTAL RESULTS PLOTTED WITH MEANS AND STANDARD DEVIATIONS

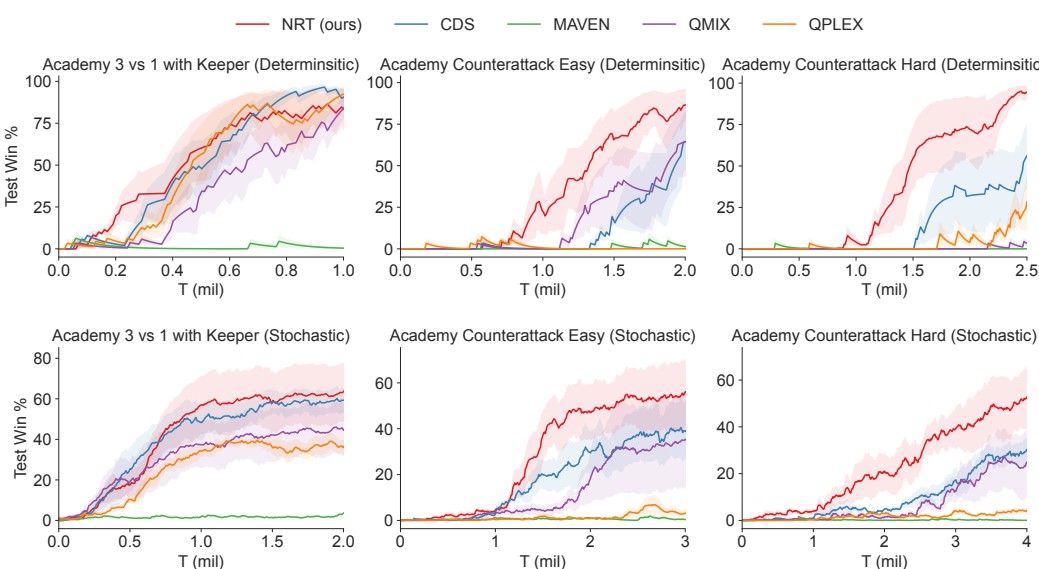

Figure 14: Comparison of our approach against baseline algorithms on GRF with deterministic environment seeds or stochastic environment seeds.

We re-plot our experimental results with means and standard deviations as shown in Fig. 14 and Fig. 15. The comparison between our algorithm and baseline is consistent with that measured with

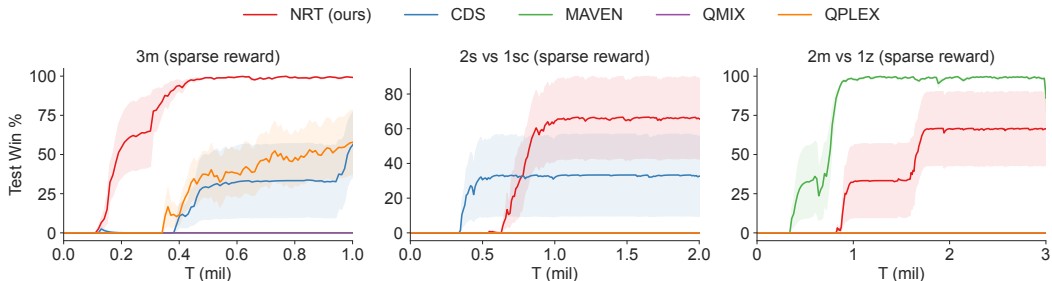

Figure 15: Comparison of our approach against baseline algorithms on SMAC with sparse rewards.

median performance. In general, our approach achieves better performance in 4 out of 6 GRF scenarios (slightly better in `Academy_3_vs_1_with_Keeper` based on the stochastic environment) and 2 out of 3 SMAC maps.

