# OpenReview forum: "Never Revisit: Continuous Exploration in Multi-Agent Reinforcement Learning"
_ICLR.cc/2023/Conference — Submitted to ICLR 2023_

### Official Review · Reviewer_hnNg · 2022-10-19

**Confidence:** 2
**Correctness:** 2
**Technical Novelty And Significance:** 2
**Empirical Novelty And Significance:** 3
**Recommendation:** 3

**Clarity, Quality, Novelty And Reproducibility:**

The biggest problem of the paper seems to be structure and language. While most sentences make sense, after rereading the paper this reviewer is unable to describe the proposed method in detail. Most terms in the background section are not explained and require in-depth knowledge of the cited papers. For example, Figure 2 is not understandable (and nearly all symbols are undefined) to a reader that has not read the original QMIX paper. While the results look good at first, the way that they are plotted makes it hard to interpret whether NRT significantly outperforms CDS, and 3 random seeds are also very little to make reliable claims in a field that is as stochastic as exploration.

**Strength And Weaknesses:**

**STRENGTH**

The problem of re-exploration is relevant, in particular for MARL, and the idea of making snapshots of the network parameters could in principle be a solution to it. The results also look promising.

**WEAKNESS**

The proposed method is not clear to this reviewer. Before a reliable review can be made, the reviewer would like to ask the authors to answer the following questions (and try to clarify them in the text as well):
1. Which role does the replay buffer play in revisitation? As long as a sample is in the replay buffer, it will be regularly updated. Does revisiting occur after a sample leaves the buffer, or for some other reason?
2. What is stored when you "store the current joint observation distribution estimator"? The output? The paramters $\xi$? And how often do you save them? Is your algorithm constant memory, or do the memory requirements grow linearly with time?
3. When you "add a new branch", which the reviewer interprets as saving (or detaching) the network parameters of the last layer, why do you expect the outputs of both the local Q-network and the mixing network to remain meaningful after the "shared modules" have been changed by gradient descent? Wouldn't any substantial change in the shared parameters require a change in the "saved" parameters as well?
4. During exploration ("execution"?), which branch is used to select actions? It seems that (eq.2) selects a branch, but what is gained when the actions are drawn from an old policy? Or do you always draw actions from the "default branch" and use (eq.2) only to modify the intrinsic reward?
5. Details on the modified intrinsic reward are hard to decipher. However, it seems that you compute a running average of a scaling factor $1+\bar\alpha$, which itself is clipped to be between $L_1$ and $L_2$. The appendix names them as usually $L_1=0.5$ and $L_2=2$, but why these values? How can the reader interpret $\bar \alpha$?
6. Wouldn't it be conceptually simpler to estimate the joint observation density of the *entire past* and scale down intrinsic rewards for observations the agent has already observed?
7. How should the 3D plots be read? What is "checkpoint time" and what is "historical point time"? Which graph would indicate a "non-revisiting" agent? If it is a front of high JS distances for new checkpoints, wouldn't a 2D-colour-plot be clearer to read?
8. Plotting the median and the variance is unorthodox and makes the results hard to interpret. Are your results statistically significant (or at least appear that way)? If you re-plot them with means and standard deviations, do the standard deviations of NRT and the best competitor overlap?

**Summary Of The Paper:**

The paper aims to improve exploration in MARL by preventing revisiting of previously explored regions. The authors propose an extension of QMIX with an arbitrary intrinsic reward module (CDS in the experiments). The proposed method NRT makes snapshots of a density estimator to detect when the distribution of joint observations changes. If a revisitation of a previously explored part of the joint observation space is detected, a snapshot of the last layer of the mixing network and the local Q-network is saved as well. Using these saved networks the authors claim they found a method to scale the out-of-the-box intrinsic reward to avoid reexploration of known areas. Empirical results are not entirely clear, but seem to show that the proposed methods improves performance.

**Summary Of The Review:**

The paper was in the current form not understandable to the reviewer. The performance of the method appears to be good, but it is not clear how and why this is, and what the downsides of the proposed methods are.

---

> ### Author Response · Authors · 2022-11-15
> **Response to Reviewer hnNg (Part I)**
>
> We thank the reviewer for your time devoted to reviewing this paper and your constructive suggestions. Here are our detailed replies.
>
> **w: The proposed method is not clear to this reviewer. Before a reliable review can be made, the reviewer would like to ask the authors to answer the following questions (and try to clarify them in the text as well)**
>
> **A for w:** We thank the reviewer for pointing out this issue. We have submitted a new version of our manuscript to make our method easier to follow. We sincerely hope the reviewer can read it and re-judge the value of our submission.
>
> **w1: Which role does the replay buffer play in revisitation? As long as a sample is in the replay buffer, it will be regularly updated. Does revisiting occur after a sample leaves the buffer, or for some other reason?**
>
> **A for w1:**  Inspired by [1], current intrinsic motivations fail to rediscover promising areas after the detachment of the previous exploration, which will cause revisitation, heavily influencing exploration efficiency. We consider that revisitation might occur according to three reasons: (1) insufficient policy representation capacity causing forgetting, (2) ineffective intrinsic motivations causing re-exploration, and (3) limited replay buffer which deletes old data. As for the role the replay buffer plays in revisitation, because (1) the revisitation will appear no matter whether a sample is in the buffer or not (as shown in Fig. 1 and Fig. 10 in the new version of our manuscript), and (2) adding the size of the replay buffer might cause unstableness (as shown in Fig. 5 in [2]), we focus the other two reasons in this paper.
>
> [1] Ecoffet, A., Huizinga, J., Lehman, J., Stanley, K. O., & Clune, J. (2019). Go-explore: a new approach for hard-exploration problems. arXiv preprint arXiv:1901.10995.
>
> [2] Hu, J., Wu, H., Harding, S. A., Jiang, S., & Liao, S. W. (2021). RIIT: Rethinking the Importance of Implementation Tricks in Multi-Agent Reinforcement Learning. arXiv preprint arXiv:2102.03479.
>
> **w2: What is stored when you "store the current joint observation distribution estimator"? The output? The parameters ξ? And how often do you save them? Is your algorithm constant memory, or do the memory requirements grow linearly with time?**
>
> **A for w2:** We will store the parameters ξ of joint observation distribution estimators every 100k time steps over the course of training. Meanwhile, we will store joint observation sequences every ten episodes in an independent buffer without deletion to adjust intrinsic motivations. The memory requirements do grow linearly with time. This is the cost we paid for optimizing the entire exploration process.
>
> **w3: When you "add a new branch", which the reviewer interprets as saving (or detaching) the network parameters of the last layer, why do you expect the outputs of both the local Q-network and the mixing network to remain meaningful after the "shared modules" have been changed by gradient descent? Wouldn't any substantial change in the shared parameters require a change in the "saved" parameters as well?**
>
> **A for w3:** We are sorry that our vague statement has caused the reviewer to misunderstand. We have reworked Section 3.2 accordingly. The augmented branches are used to provide sufficient policy representation capacity for the sub-space where revisitation has already happened. They are still trained while being activated during training. Our ablation study in Section 5.1 has demonstrated that our dynamically augmented neural network structure is crucial in complex environments such as Academy Counterattack Hard under the stochastic setting.
>
> **w4: During exploration ("execution"?), which branch is used to select actions? It seems that (eq.2) selects a branch, but what is gained when the actions are drawn from an old policy? Or do you always draw actions from the "default branch" and use (eq.2) only to modify the intrinsic reward?**
>
> **A for w4:** We supplemented more details on how to select branches during sampling and training in the new version of our manuscript. Based on our formal definition of revisitation, we introduce an associated joint observation set for each revisitation tuple as shown in Eq.3 (in the new version of our manuscript). However, during decentralized execution, one agent can only obtain its own observation. Thus, we introduce the marginal condition based on Eq. 3 to guide each agent to select local-Q branches for both sampling and training (Eq.4 in the new version of our manuscript). As discussed in 'A for w3', there are no old policies during sampling or training.

---

> > ### Author Response · Authors · 2022-11-15
> > **Response to Reviewer hnNg (Part II)**
> >
> > **w5.1: Details on the modified intrinsic reward are hard to decipher. However, it seems that you compute a running average of a scaling factor $1+\bar{\alpha}$, which itself is clipped to be between L1 and L2. The appendix names them as usually L1 = 0.5 and L2 = 2, but why these values?**
> >
> > **A for w5.1:** We have revised Section 3.3 to make it easier to follow. The value of L1 is roughly searched in {0.5, 1.0}, while the value of L2 is roughly searched in {2.0, 4.0}.
> >
> > **w5.2: How can the reader interpret $\bar{\alpha}$?**
> >
> > **A for w5.2:** We use the novelty of joint observations, which is measured by Euclidean distance, to adjust intrinsic rewards, which are calculated according to the information theory. The measure of these two items needs to be carefully checked. In this paper, we keep intrinsic rewards' measurement unchanged, while using the running average of joint observations' novelty ($\bar{\alpha}$) to multiply or shrink intrinsic rewards as inspired by [3]
> >
> > [3] Badia, A. P., Sprechmann, P., Vitvitskyi, A., Guo, D., Piot, B., Kapturowski, S., ... & Blundell, C. (2020). Never give up: Learning directed exploration strategies. arXiv preprint arXiv:2002.06038.
> >
> > **w6: Wouldn't it be conceptually simpler to estimate the joint observation density of the entire past and scale down intrinsic rewards for observations the agent has already observed?**
> >
> > **A for w6:** In this paper, we first want our community to notice the existence of learning policies' periodical revisitation based on regular intrinsic motivations and realize its harm to exploration efficiency. Thus we store joint observation distributions periodically and calculated the JS distance between them to analyze the similarity between policies learned in different periods. After that, we introduce our algorithms to solve it and achieve good performance. We thank the reviewer for this valuable advice and will research how to simplify our algorithm in future work.
> >
> > **w7.1: How should the 3D plots be read? What is checkpoint time and what is historical point time?**
> >
> > **A for w7.1:** We have supplemented a detailed explanation of labels in Figure 1 for better clarification. During learning, we will store the joint observation distribution induced by the joint policy π every 100k time steps, named historical point time. Meanwhile, for every 100k time step, we will calculate the JS distance between the current distribution and all historical points, named checkpoint time.
> >
> > **w7.2: Which graph would indicate a non-revisiting agent?**
> >
> > **A for w7.2:** We consider the right one in Figure 1 can be regarded as a continuous exploration until the optimal strategy is explored, because the JS distances are large and stable compared with baselines.
> >
> > **w8: Plotting the median and the variance is unorthodox and makes the results hard to interpret. Are your results statistically significant (or at least appear that way)? If you re-plot them with means and standard deviations, do the standard deviations of NRT and the best competitor overlap?**
> >
> > **A for w8:** We think plotting the median can show more information. Meanwhile, this method is also commonly used, such as [4]. The shaded areas shown in our experiments are the scale of means plus or minus standard deviations. The standard deviations of NRT and the best competitor might overlap. However, the means of NRT still outperform baselines in most environments.
> >
> > [4] Samvelyan, M., Rashid, T., De Witt, C. S., Farquhar, G., Nardelli, N., Rudner, T. G., ... & Whiteson, S. (2019). The starcraft multi-agent challenge. arXiv preprint arXiv:1902.04043.

---

> > > ### Comment · Reviewer_hnNg · 2022-11-16
> > > **Please clarify further**
> > >
> > > Thanks to the authors for their answers. Could you please answer the questions below, where I did not fully understood your answer?
> > >
> > > w1: the observation that revisitation is independent of the replay buffer is interesting and should be empirically demonstrated. However, the other possibilities should also be excluded t make a good argument: (1) any policy that is updated using a replay buffer (but not sampling methods like PPO) *cannot forget* samples that are still in the buffer. If the representational capacity is too low, all regions will be equally bad approximated. (2) it should be relatively easy to evaluate whether the intrinsic motivation *method* is to blame by using explicit counts in toy domains. (3) this possibility can be evaluated in toy domains, too.
> > >
> > > w2: thank you for your answer.
> > >
> > > w3: I don't understand this answer. What is a "branch"? Are they additional network heads you add over time? With which loss are they trained? What is their role in avoiding revisitation? In Figure 2 (which I still do not understand) it looks as if a branch is additional hyper-layer W2 that is connected using a $L_1$ norm (what does this mean?) and a "default switch". Please elaborate how this works, ideally by providing the forward equations for the above unclear parts and the losses they get trained with. Also please indicate which parts (branches) are used during exploration and/or execution.
> > >
> > > w4: see w3
> > >
> > > w5.1: but why these values?
> > >
> > > w5.2: so $\bar \alpha$ should be interpreted as "joint observation novelty", right?
> > >
> > > w6: thank you for your answer.
> > >
> > > w7: thank you for the clarification. However, I am still at a loss how I should interpret these figures. I observe that (i) NRT has a larger (and earlier) coverage rate, which I assume is measured independently of the 3D plot; (ii) the 3D plot of NRT is smoother. However, I still do not understand what a "good" figure would look like and why e.g. in Figure 5 the plotted graph seems to form an arc in x-y space, which should be a line according to your description. Maybe a 2D-color plot would be clearer to read!
> > >
> > > w8: I disagree with this conclusion. Overlapping standard deviations are a sign that individual measurements of the two distributions can not be distinguished statistically. However, you could see if the standard errors overlap to make a statement about whether the *means* of the distribution can be distinguished. Please refrain from making statements like "NRT still outperform baselines in most environments", unless you can make those with some kind of statistical significance. You can always see a "trend in the data" or a "tendency to perform better", but these are different from statements that one can expect to hold when the experiments are replicated.

---

> > > > ### Author Response · Authors · 2022-11-17
> > > > **Thanks for further feedback! (Part II)**
> > > >
> > > > **w3.2: What is their role in avoiding revisitation?**
> > > >
> > > > **A for w3.2:** Branches are used to construct dynamic augmentation of neural networks according to the degree of exploration for providing sufficient representational capacity.
> > > >
> > > > **w3.3: which parts (branches) are used during exploration and/or execution.**
> > > >
> > > > **A for w3.3:** According to your advice, we provide the pseudo-code of NRT’s sampling process as shown in Algorithm 1 in Section B.1.
> > > >
> > > > Based on each revisitation, we separate the whole joint observation space into sub-spaces with Eq. 3. However, during decentralized execution, one agent can only obtain its own observation. Thus, we introduce the marginal condition Eq. 4 based on Eq. 3 to guide each agent to select local-Q branches for both sampling and training. For agent $i \in I$, if its local observation is in one revisitation's individual observation set (Eq.4), the related augmented local-Q branch will be activated.
> > > >
> > > > This brings two questions:
> > > >
> > > > (1) How to switch augmented local-Q branches when $o_i$ belongs to several sets simultaneously? In this paper, we will check according to the created order and activate the earliest created branch.
> > > >
> > > > (2) How do we select mixing network branches when agents select different local-Q branches? In this paper, each augmented mixing network branch will be activated only when more than half of the agents choose the corresponding augmented local-Q branch.
> > > >
> > > > Our neural networks will switch to the default branch if none of the augmented branches is activated or created.
> > > >
> > > > **w3.4: With which loss are they trained?**
> > > >
> > > > **A for w3.4:** Each branch is trained when it is activated.
> > > >
> > > > The original QMIX train $Q_{tot}$ with
> > > >
> > > > $$\mathcal{L}(\theta)=\sum_{i=1}^{b}\left[\left(y_{i}^{t o t}-Q_{t o t}(\boldsymbol{\tau}, \mathbf{u}, s ; \theta)\right)^{2}\right]$$
> > > >
> > > > where $y^{t o t}=r+\gamma \max_{\mathbf{u}^{\prime}}Q_{t o t}\left(\boldsymbol{\tau}^{\prime}, \mathbf{u}^{\prime}, s^{\prime} ; \theta^{-}\right)$, $\theta^{-}$ are the parameters of a target network as in DQN, and $b$ is the batch size of transitions sampled from the replay buffer.
> > > >
> > > > In this paper, we train $Q_{tot}$ with
> > > >
> > > > $$\mathcal{L}(\theta)=\sum_{i=1}^{b}\left[\left(y_{i}^{t o t}-Q_{t o t}(\boldsymbol{\tau}, \mathbf{u}, s ; \theta)\right)^{2}\right] + \sum_{i=1}^{b} ||Q_{tot} ^ {I, -}||_1$$
> > > >
> > > > where $Q_{tot} ^ {I, -} = (\mathtt{stopgrad}([Q_1, ..., Q_n] W_1 + b_1)) W_B^T + b_B$ is an additional regularization loss item. stop-grad is used to stop gradient back-propagation. Thus we can only regularize parameters related to each mixing network branch.
> > > >
> > > > **w5.1: but why these values?**
> > > >
> > > > **A for w5.1:** We intuitively want the dynamic weight of intrinsic motivations not to be too high or too low. Thus we introduce a clip function while calculating it. We roughly adjust L1 and L2 ( L1 $\in$ {0.5, 1.0} and L2 $\in$ {2.0, 4.0}, which are intuitively chosen) and get good results.
> > > >
> > > > **w5.2:  interpreted as "joint observation novelty", right?**
> > > >
> > > > **A for w5.1:** Right, $\bar{\alpha}$ can be interpreted as joint observation novelty which is normalized across the whole learning period.
> > > >
> > > > **w7: A 2D-color plot would be clearer to read!**
> > > >
> > > > **A for w7:** According to your advice, we further provide a 2D-Color plot of Figure 1's JS distances as shown in Figure 11 and Figure 12 in Section C.2.
> > > >
> > > > **w8: I disagree with this conclusion. Overlapping standard deviations are a sign that individual measurements of the two distributions can not be distinguished statistically. However, you could see if the standard errors overlap to make a statement about whether the means of the distribution can be distinguished. Please refrain from making statements like "NRT still outperform baselines in most environments", unless you can make those with some kind of statistical significance. You can always see a "trend in the data" or a "tendency to perform better", but these are different from statements that one can expect to hold when the experiments are replicated.**
> > > >
> > > > **A for w8:** According to your advice, we further re-plot our experimental results with means and standard deviations as shown in Fig. 14 and Fig. 15. in Section C.4. The comparison between our algorithm and baseline is consistent with that measured with median performance. In general, our approach achieves better performance in 4 out of 6 GRF scenarios (slightly better in Academy 3 vs 1 with Keeper based on the stochastic environment) and 2 out of 3 SMAC maps.

---

> > > > > ### Comment · Reviewer_hnNg · 2022-11-21
> > > > > **Last question**
> > > > >
> > > > > Thank you for your clarifications. However, my original question seems to still be valid:
> > > > > - according to Alg.1 you sample one "branch" at a time during execution.
> > > > > - according to (eq.9) you update $Q_{tot}$ (which contains both shared and branch-specific parameters) with the default $L_2$ QMIX loss and the branch-specific parameters with an $L_1$ loss on $Q^I_{tot}$ that effectively aims to minimize the output of $Q^I_{tot}$.
> > > > >
> > > > > So which "branch"'s parameters are used for the training of (eq.9)?
> > > > > - this choice/selection must be clear from the equations (which it currently is not).
> > > > > - if it is a single "branch" selected by Alg.1, then what happens with the other branches when the shared parameters change?
> > > > > - or do you select one branch per mini-batch sample? If so, are you relying on mini-batches that contain samples from all branches? If so, wouldn't it make more sense to store the samples from different branches in different replay buffers to guarantee this property?
> > > > > - however the "branch" of a specific training sample is selected, how does this selection change the usual QMIX loss (besides changing $Q^I_{tot}$)? Is the intrinsic reward computation different? Otherwise all "branch"es would converge towards the same values, but the current text reads as if "$r^I$ can be any intrinsic reward".
> > > > > - How is $Q^I_{tot}$ initialized for a new branch? Figure 2 indicates that a new branch is "initialized with the latest default branch", but doesn't this simply mean that $Q^I_{tot}\approx 0$?

---

> > > > > > ### Author Response · Authors · 2022-11-22
> > > > > > **Thanks for further feedback!**
> > > > > >
> > > > > > We thank the reviewer for the provided comments.
> > > > > >
> > > > > > In general, we use augmented branches to achieve a dynamically increasing representational capacity as inspired by [1]’s functionally compositional structures. They are quite necessary for complex tasks as shown in our ablation study.
> > > > > >
> > > > > > **So which "branch"'s parameters are used for the training?**
> > > > > >
> > > > > > According to each revisitation, we separate the whole joint observation space based on Eq.3. If one joint observation $\boldsymbol{o}$ is in $\Omega_{b}$ related to b th revisitation, b th augmented branch will be activated for estimating $Q_{tot}(\boldsymbol{o})$. If no revisitation has been detected or $\boldsymbol{o}$ relates to no revisitation, the default branch will be activated.
> > > > > >
> > > > > > For one trajectory, its different joint observations are related to various branches. Thus all branches will be activated and updated during training based on per mini-batch sample of trajectories.
> > > > > >
> > > > > > As for how to conform to the CTDE framework and other details, we have discussed it in Section 3.2.
> > > > > >
> > > > > > **What happens with the other branches when the shared parameters change?**
> > > > > >
> > > > > > In environments we currently consider, revisitation will only happen once or twice based on our approach. Meanwhile, all branches will be activated and updated during training based on per mini-batch sample of trajectories. Shared parameters' changes are followed by all branches during training. Thus the mismatch caused by shared parameters' changes rarely occurs.
> > > > > >
> > > > > > For extremely complex tasks or environments, revisitation might occur numerous times followed by numerous branches. In this condition, several branches might rarely be activated during training, causing serious mismatches when activated. We will study this issue in future works if we discover it. Thanks for your valuable comments.
> > > > > >
> > > > > > **store the samples from different branches in different replay buffers**
> > > > > >
> > > > > > As mentioned above,  for one trajectory, its different joint observations are related to various branches. Meanwhile, following most algorithms in the multi-agent setting, we use recurrent neural networks and will sample full trajectories per mini-batch to train them. Thus we cannot store joint observations from different branches in different replay buffers.
> > > > > >
> > > > > > **how does this selection change the usual QMIX loss**
> > > > > >
> > > > > > The selection of branches is based on the detection of revisitation. Meanwhile, intrinsic rewards are also adjusted according to the detection of revisitation. This relationship will change the usual QMIX loss. Meanwhile, one joint observation only corresponds to one branch, so branches cannot be exactly the same after learning.
> > > > > >
> > > > > > **how is** $Q_{t o t}^{I}$ **initialized for a new branch?**
> > > > > >
> > > > > > After each detection of revisitation, we will separate the joint observation space corresponding to the default branch according to Eq.3, and add new branches activated in the separated area. For avoiding degradation of performance, we initialize new branches with the latest default branch.
> > > > > >
> > > > > > Based on the L1 regularization added to mixing network branches, $Q_{t o t}^{I} \approx 0$. However, it still can provide additional representation capacity when necessary, that is what we expect.
> > > > > >
> > > > > > [1] Mendez, J. A., van Seijen, H., & Eaton, E. (2022). Modular lifelong reinforcement learning via neural composition. arXiv preprint arXiv:2207.00429.

---

> > > > ### Author Response · Authors · 2022-11-17
> > > > **Thanks for further feedback! (Part I)**
> > > >
> > > > We thank the reviewer for your time devoted to this paper. We also appreciate the valuable comments, which helped us improve the paper's strengths significantly.
> > > >
> > > > **w1.1: any policy that is updated using a replay buffer (but not sampling methods like PPO) cannot forget samples that are still in the buffer. If the representational capacity is too low, all regions will be equally bad approximated.**
> > > >
> > > > **A for w1.1:** As the reviewer pointed out, if the representational capacity is too low, all regions will be equally bad approximated. Thus we dynamically augment our neural network structure along with exploration for sufficient representational capacity. Our ablation study in Section 5.1 has demonstrated that constructing dynamic augmentation of neural networks according to the degree of exploration is crucial in complex environments.
> > > >
> > > > **w1.2: it should be relatively easy to evaluate whether the intrinsic motivation method is to blame by using explicit counts in toy domains.**
> > > >
> > > > **A for w1.2:** With explicit counts in a UCB (Upper Confidence Bound) style, agents can explore the optimal strategy in our toy and discrete maze domain within 1M time steps. Unfortunately, count-based rewards are very difficult to use in complex environments such as SMAC and GRF, whose joint observation space is a continuous space with more than 100 dimensions. That's why the community tends to introduce intrinsic motivations. However, in this paper, we discover that coming with intrinsic rewards is the issue of revisitation – the relative values of intrinsic rewards, estimated based on neural networks, fluctuate during learning, causing failures of rediscovering promising areas after detachment of exploration. We reveal this dangerous phenomenon and propose at least a preliminary solution.
> > > >
> > > > **w1.3: this possibility can be evaluated in toy domains, too.**
> > > >
> > > > **A for w1.3:** According to your advice, we further demonstrate that the size of replay buffer has trivial influence on revisitaion in Section C.3.  We still use CDS as an example of intrinsic motivations and test it in our maze environment. Figure 13 shows the interval of periodical revisitation is not determined by the size of replay buffer.
> > > >
> > > > **w3:  I don't understand this answer.**
> > > >
> > > > **A for w3:** According to your advice, we further provide details of Figure 2 in Sec. B.1. Here are our detailed replies.
> > > >
> > > > **w3.1: What is a "branch"?**
> > > >
> > > > **A for w3.1:** Based on each revisitation, we separate the whole joint observation space into sub-spaces with Eq. 3. To provide sufficient policy representation capacity, we dynamically augment neural networks for each sub-space as shown in Fig. 2.
> > > >
> > > > The last layers of agents’ local Q-networks are independent across sub-spaces, while the previous layers are shared for benefits in learning efficiency from parameter sharing. Here we name the augmented last layer of agents’ local Q-networks as the **augmented local-Q branch** activated in the sub-space of related revisitation. The original last layer of agents’ local Q-networks initialized at the beginning of learning is named as **default local-Q branch** activated when no revisitation is detected or in the sub-space related to no revisitation.
> > > >
> > > > For achieving a sufficient representational capacity of the mixing network, we augment it along with augmentations of local Q-networks as shown in Fig. 2.
> > > >
> > > > The original QMIX calculates $Q_{tot}$ with:
> > > >
> > > > $$Q_{tot} = ([Q_1, ..., Q_n] W_1 + b_1) W_2^T + b_2$$
> > > >
> > > > where $Q_i$ is the local Q-value of agent $i \in I$, $W_1$ is a $n \times m$ matrix, $b_1$ and $W_2$ are vectors with $m$ dimensions, $b_2$ is a variable with 1 dimension. $W_1$ and $W_2$ are both with non-negative values. $W_1$, $W_2$, $b_1$, and $b_2$ are all outputs of several hyper networks with the state as input.
> > > >
> > > > However, the mixing network might not have enough capacity while the number of local-Q branches increases. Thus we further expand the mixing network. Our QMIX-style mixing network is
> > > >
> > > > $$Q_{tot} = ([Q_1, ..., Q_n] W_1 + b_1) (W_2^T + W_B^T) + b_2 + b_B = (([Q_1, ..., Q_n] W_1 + b_1) W_2^T + b_2) + (([Q_1, ..., Q_n] W_1 + b_1) W_B^T + b_B) = Q_{tot}^S + Q_{tot}^I$$
> > > >
> > > > where $W_B$ is a vector with $m$ dimensions and $b_B$ is a variable with 1 dimension.
> > > >
> > > > $Q_{tot}^S$ is shared among sub-spaces with shared parameters of $W_1$, $b_1$, $W_2$ and $b_2$. $Q_{tot}^I$ is an individual part of each sub-space's mixing network with individual parameters of $W_B$ and $b_B$. Here we name the augmented $W_B$ and $b_B$ related to each revisitation as the **augmented mixing network branch**. The original $W_B$ and $b_B$ initialized at the beginning of learning are named as **default mixing network branch** activated when no revisitation is detected or the sub-space is related to no revisitation.

---

### Official Review · Reviewer_iFpg · 2022-10-22

**Confidence:** 4
**Correctness:** 2
**Technical Novelty And Significance:** 3
**Empirical Novelty And Significance:** 3
**Recommendation:** 5

**Clarity, Quality, Novelty And Reproducibility:**

# Clarity
Discussed in the "Strengths and Weaknesses" section. In summary, the problem setting is clear; however, the decisions behind the design of the methods are not.

# Quality
The results are good; however, the methods may be seen as a bit heuristic-based without enough justification for the heuristics.

# Novelty
The paper addresses a novel and important problem.

# Reproducibility
No code is provided and the paper is missing details to reproduce effectively (e.g. how to find revisitation tuples, how branched Q-networks are trained, etc.)

**Strength And Weaknesses:**

# Strengths
* The challenge of revisitation is well motivated.
* The proposed method seems effective for avoiding the revisitation issue.
* Evaluation is performed on the Google Research Football setting which is still challenging for many cooperative MARL methods.

# Weaknesses
* It is not clear from the writing whether revisitation is uniquely an issue in multi-agent problems. I could imagine that learning decentralized policies accentuates this issue, but it should be explicitly motivated.
* The design decisions in the methods section (specifically revolving around the branching architecture) could be motivated better (more details in questions below).
* The approach seems as though it would scale poorly, as it requires storing new parameters for each "revisitation tuple" detected. Furthermore, the computation of intrinsic rewards scales with the number of revisitation tuples.
* The method's generality would be more convincing if also tested on the standard StarCraft benchmarks in addition to GRF.

# Questions
* Doesn't computing revisitation tuples require storing all transitions seen over the course of training? And naively it would take $O(T^2)$ time since you have to iterate over $t'$ and $t''$. The details on this are a bit sparse: how do you overcome these challenges? I assume some sort of sub-sampling is involved.
* It is not clear why branching the Q-function is useful for avoiding revisitation. Why would re-using old policies help avoid revisiting previously seen sub-spaces? Intuitively it would do the opposite. Is the point that you can use these branches to "absorb" bad data from thoroughly explored states that come from your replay buffer such that the new branches are only trained on data from novel sub-spaces? If so, perhaps a simpler filtering or re-weighting scheme scheme could be similarly effective and less computationally expensive.
* How are the branched functions trained? Do you update different ones depending on whether they meet the criteria in Eqn 2? Perhaps more explanation here will address the question in the previous bullet as well.

**Summary Of The Paper:**

The authors propose a novel multi-agent architecture and intrinsic reward scheme for avoiding the issue of revisiting previously seen states in multi-agent reinforcement learning. They demonstrate improved performance in a didactic domain as well as in Google Research Football.

**Summary Of The Review:**

This work achieves impressive performance on a challenging domain by addressing a well-motivated problem; however, a significant portion of the methodological decisions (i.e. branching architecture) are lacking convincing motivation and important details. At this time I cannot recommend acceptance; however, I am willing to revise my opinion if the clarity of section 3.2 is significantly improved (in terms of details regarding how branching architectures factor into training and execution as well as motivation for why they are necessary).

---

> ### Author Response · Authors · 2022-11-15
> **Response to Reviewer iFpg (Part I)**
>
> Thanks for finding our work novel. We provide clarification to the Reviewer's comments in the following.
>
> **w1: It is not clear from the writing whether revisitation is uniquely an issue in multi-agent problems. I could imagine that learning decentralized policies accentuates this issue, but it should be explicitly motivated.**
>
> **A for w1:** Inspired by [1], revisitation after the detachment of exploration might be a common issue coming with intrinsic motivations. In multi-agent systems, this issue will be enlarged due to its exponentially growing search space according to the number of agents. Because the authors focus on the field of multi-agent systems, we research from the perspective of exploration problems in multi-agent.
>
> [1] Adrien Ecoffet, Joost Huizinga, Joel Lehman, Kenneth O Stanley, and Jeff Clune. Go-explore: a new approach for hard-exploration problems. arXiv preprint arXiv:1901.10995, 2019.
>
> **w2: The design decisions in the methods section (specifically revolving around the branching architecture) could be motivated better (more details in questions below).**
>
> **A for w2:** We thank the reviewer for pointing out this issue. We have submitted a new version of our manuscript to make our method easier to follow. We sincerely hope the reviewer can read it and re-judge the value of our submission.
>
> **Q1: Doesn't computing revisitation tuples require storing all transitions seen over the course of training? And naively it would take O(T^2)
>  time since you have to iterate over t′and t″. The details on this are a bit sparse: how do you overcome these challenges? I assume some sort of sub-sampling is involved.**
>
> **A for Q1:** Computing revisitation tuples indeed requires storing all transitions seen over the course of training. This is the cost we paid for optimizing the entire exploration process. For time efficiency, our historical points and checkpoints are introduced every 100k time steps. Meanwhile, we store joint observation sequences every ten episodes and form a sort of sub-sampling while adjusting intrinsic motivations.
>
> **Q2.1: It is not clear why branching the Q-function is useful for avoiding revisitation.**
>
> **A for Q2.1:** Our dynamic augmentation neural network structure is introduced for providing sufficient policy representation along with continuous exploration. Our ablation study based on GRF demonstrates its significance in complex environments.
>
> **Q2.2: Why would re-using old policies help avoid revisiting previously seen sub-spaces?**
>
> **A for Q2.2:** We are sorry that our vague statement has caused the reviewer to misunderstand. We do not reuse old policies. Based on each revisitation, we separate the whole joint observation space into sub-spaces with Eq. 3. To provide sufficient policy representation capacity, we dynamically augment the neural network structure for each sub-space as shown in Fig. 2.
>
> **Q2.3: Intuitively it would do the opposite. Is the point that you can use these branches to "absorb" bad data from thoroughly explored states that come from your replay buffer such that the new branches are only trained on data from novel sub-spaces? If so, perhaps a simpler filtering or re-weighting scheme could be similarly effective and less computationally expensive.**
>
> **A for Q2.3:** In this paper, we choose to dynamically increase representational capacity to store more and more knowledge along with continuous exploration. How to use a simpler filtering or re-weighting scheme for less computationally expensive might be our future study. We thank the reviewer for pointing out this idea.
>
> **Q3: How are the branched functions trained? Do you update different ones depending on whether they meet the criteria in Eqn 2? Perhaps more explanation here will address the question in the previous bullet as well.**
>
> **A for Q3:** We have added more details about our method in the new version (Section 3.2 and Section B.1). The local-Q branches are activated according to Algorithm 1 in Section B.1 and are updated while being selected during training.
>
> Here are detailed processes. If no augmented local-Q branches are created, agents will choose the default local-Q branch. If there are augmented local-Q branches, each agent $i \in I$ will check whether its local observation $o_i$ is related to any augmented branch based on Eq.4. If $o_i$ is in only one marginal set $\Omega_{b}^{i}$, agent $i$ will activate $b$ th augmented branch. If more than one augmented branch is going to be activated, the earliest created one will be activated. If no augmented branch is going to be activated, the default branch will be activated.
>
> As for augmented mixing network branches, they will be activated and updated only when more than half of agents choose the corresponding augmented local-Q branch.

---

> > ### Author Response · Authors · 2022-11-15
> > **Response to Reviewer iFpg (Part II)**
> >
> > **w3: The approach seems as though it would scale poorly, as it requires storing new parameters for each "revisitation tuple" detected. Furthermore, the computation of intrinsic rewards scales with the number of revisitation tuples.**
> >
> > **A for w3:** The scale of our neural network parameters will indeed increase along with the detection of revisitation for achieving continuous exploration. However, based on our modules on intrinsic motivations adjustment, the number of revisitation is controlled at a very small level (1 or 2) as shown in Fig.1, Fig.5 (a), Fig. 10, and Fig. 12 in our manuscript's new version.
> >
> > **w4: The method's generality would be more convincing if also tested on the standard StarCraft benchmarks in addition to GRF.**
> >
> > **A for w4:** We further compare our approach with baselines in the StarCraft II micromanagement challenge (SMAC) benchmark based on sparse rewards (in Sec.5.2). Previous work has achieved remarkable performance in many challenging SMAC tasks. However, their success heavily relays on dense rewards provided by the environment. Based on sparse rewards that only occur at the end of one episode, baselines such as vanilla QMIX can not win even in easy maps, while our approach achieves remarkable performance. We hope that the additional experimental results on SMAC can provide more support to demonstrate the significance of our approach.

---

> ### Author Response · Authors · 2022-11-17
> **Looking forward to further discussions!**
>
> Dear reviewer,
>
> We were wondering if our response and revision have resolved your concerns. In our responses and updated manuscripts, we gave a more detailed explanation of our method and we included more experiments. Please let us know if you have any further concerns, and we are looking forward to more discussions to improve our manuscript further.
>
> Best regards,
>
> The Authors

---

### Official Review · Reviewer_2T1q · 2022-10-24

**Confidence:** 4
**Correctness:** 3
**Technical Novelty And Significance:** 2
**Empirical Novelty And Significance:** 2
**Recommendation:** 5

**Clarity, Quality, Novelty And Reproducibility:**

### Clarity:
The approach section is unclear and not rigorous. The reviewer found it hard to follow. Please see the weakness section for more details.

### Quality:
The claims in the paper are partially supported by the experiments. Experimental results on more complex tasks would make it more convincing.

### Novelty:
The reviewer found the identified ‘revisitation’ issue an important problem. However, more clarification on the technical content is needed.

### Reproducibility:
No code is provided and the technical content is not very clear. The reviewer thinks the results may be difficult to reproduce.


**Strength And Weaknesses:**

### Strength:

The paper is well-motivated and the problem it aimed to address is important.  Specifically,  the paper identified and formally defined the ‘revisitation’ issue. The reviewer thinks it is an important step for our community to better address the issue.

### Weakness:

While the motivation and the introduction is clear, the reviewer has some concerns regarding the rigorousness of the presentation of the technical content and the generality of the experimental results. Please see the detailed comments below:

1. The reviewer found Sec 3.2 difficult to follow. Many terms are either undefined or vaguely defined.

    - In p.4, the paper reads ‘augment the mixing network in the CTDE framework to achieve diverse credit assignments across agents’. It is unclear what ‘diverse credit assignments’ is referring to. A formal definition would be helpful.

   - In p.4, the paper reads ‘The red item added to the independent part of each class is an L1 regularization term for filtering out useless diversity.’ Both ‘red item’ and ‘the independent part’ are undefined in the text. In addition, it is unclear to the reviewer what is ’useless diversity’ referring to.

   - In p.4 the paper mentioned ‘According to each agent’s independent observation, the first class satisfying the marginal condition in Eq. 2 …’ Doesn’t Eq.2 consider joint observation?

   - It is unclear to the reviewer how Eq (2) helps select a branch that prevents revisitation. Could you elaborate?

2. While the proposed approach outperformed baselines in some tasks of GRF, the reviewer found reporting results on only three tasks not convincing. Particularly, the three selected tasks involved a very limited number of controlled agents. It is unclear if the proposed approach could be applied to more complex settings. Reporting results in GRF 11 vs 11 full game or in StarCraft II micromanagement challenge (SMAC) would be very helpful.


3. The term ‘sub-space’ is used throughout the paper without formal definition.


4. Figure 1 reports the ‘coverage rate’ of baselines and the proposed approach. It is unclear to the reviewer how the ‘coverage rate’ is computed. Without the definition of the metrics, it is hard to interpret the results.


**Summary Of The Paper:**

This paper studies multi-agent exploration, which is an important problem for more efficient multi-agent reinforcement learning. The authors first pointed out that the issue of ‘revisition’ hurts the exploration and learning efficiency of  existing intrinsic exploration methods. To address the issue, the author ​​proposed to add branches to agents’ local Q-networks and the mixing network to prevent receiving repeated observations. In addition, auxiliary intrinsic rewards are proposed to further avoid revisitation. The proposed approach is evaluated on three academy tasks of Google research Football (GRF). The results show that the proposed approach outperforms baselines.

**Summary Of The Review:**

In summary, the paper identified and addressed the important ‘revisitation’ problem. However, the presentation of technical content is unclear and the experimental results are not very convincing. Major revision is needed.

---

> ### Author Response · Authors · 2022-11-15
> **Response to Reviewer 2T1q**
>
> Thanks for finding our work important. We provide clarification to the Reviewer's comments in the following.
>
> **w1: The reviewer found Sec 3.2 difficult to follow. Many terms are either undefined or vaguely defined.**
>
> **A for w1:** We thank the reviewer for pointing out this issue. We have submitted a new version of our manuscript to make our method easier to follow with a major revision. We sincerely hope the reviewer can read it and re-judge the value of our submission.
>
> **w1.1: In p.4, the paper reads ‘augment the mixing network in the CTDE framework to achieve diverse credit assignments across agents’. It is unclear what ‘diverse credit assignments’ is referring to. A formal definition would be helpful.**
>
> **A for w1.1:** We consider the mixing network can assign credit across agents for achieving coordination. In this paper, for handling revisitation, we dynamically augment our neural network structure to provide sufficient policy representation capacity. Along with augmented local Q-networks, we augment the mixing network simultaneously to provide the possibility to achieve various credit assignments across agents in different branches.
>
> **w1.2: In p.4, the paper reads ‘The red item added to the independent part of each class is an L1 regularization term for filtering out useless diversity.’ Both ‘red item’ and ‘the independent part’ are undefined in the text. In addition, it is unclear to the reviewer what is ’useless diversity’ referring to.**
>
> **A for w1.2:** We have reorganized our manuscript to make our method clear. We thank the reviewer for pointing out our ambiguous expression. Because our whole trajectory is spliced from different branches, we expect credit assignments across agents to not differ too much for stability in coordination. Thus we add an L1 regularization term to each mixing network branch to encourage them to work only when necessary.
>
> **w1.3: In p.4 the paper mentioned ‘According to each agent’s independent observation, the first class satisfying the marginal condition in Eq. 2 …’ Doesn’t Eq.2 consider joint observation?**
>
> **A for w1.3:** Following our Definition 1, we introduce the associated joint observation set for each branch. However, during decentralized execution, one agent can only obtain its own observation. Thus, we introduce the marginal condition based on the associated joint observation set to guide each agent to select local-Q branches for both sampling and training (Eq. 4 in the new version). In the new version of our manuscript, we further use Algorithm 1 in Section B.1 to show the sampling process of our approach.
>
> **w1.4: It is unclear to the reviewer how Eq (2) helps select a branch that prevents revisitation. Could you elaborate?**
>
> **A for w1.4:** Our dynamic augmentation neural network structure is introduced for providing sufficient policy representation along with continuous exploration. Our ablation study based on GRF demonstrates its significance in complex environments.
>
> **w2: While the proposed approach outperformed baselines in some tasks of GRF, the reviewer found reporting results on only three tasks not convincing. Particularly, the three selected tasks involved a very limited number of controlled agents. It is unclear if the proposed approach could be applied to more complex settings. Reporting results in GRF 11 vs 11 full game or in StarCraft II micromanagement challenge (SMAC) would be very helpful.**
>
> **A for w2:** We further compare our approach with baselines in the StarCraft II micromanagement challenge (SMAC) benchmark based on sparse rewards (in Sec.5.2). Previous work has achieved remarkable performance in many challenging SMAC tasks. However, their success heavily relays on dense rewards provided by the environment. Based on sparse rewards that only occur at the end of one episode, baselines such as vanilla QMIX can not win even in easy maps, while our approach achieves remarkable performance. We hope that the additional experimental results on SMAC can provide more support to demonstrate the significance of our approach.
>
>  **w3: The term ‘sub-space’ is used throughout the paper without formal definition.**
>
> **A for w3:** The term ‘sub-space’ is used to define one subset of the joint observation space. According to your advice, we further add a definition of it in our new manuscript (Section B.1), where we heavily use the term 'sub-space'.
>
> **w4: Figure 1 reports the ‘coverage rate’ of baselines and the proposed approach. It is unclear to the reviewer how the ‘coverage rate’ is computed. Without the definition of the metrics, it is hard to interpret the results.**
>
> **A for w4:** We have re-emphasized the meaning of ‘coverage rate’ in the paper, which is calculated based on the whole joint observation space. Because our toy maze environment has a discrete joint observation space, we can explicitly count it. If one joint observation is visited, we count it as covered.

---

> ### Author Response · Authors · 2022-11-17
> **Looking forward to further discussions!**
>
> Dear reviewer,
>
> We were wondering if our response and revision have resolved your concerns. In our responses and updated manuscripts, we gave a more detailed explanation of our method and we included more experiments. Please let us know if you have any further concerns, and we are looking forward to more discussions to improve our manuscript further.
>
> Best regards,
>
> The Authors

---

### Official Review · Reviewer_XHpU · 2022-10-26

**Confidence:** 2
**Correctness:** 3
**Technical Novelty And Significance:** 2
**Empirical Novelty And Significance:** 2
**Recommendation:** 3

**Clarity, Quality, Novelty And Reproducibility:**

In its current state the paper is far from being self contained and is difficult to read.
Because of that it is also no possible to evaluate the novelty and reproducibility of the paper.

**Strength And Weaknesses:**

Strengths:

_ The paper presents decent results on Google research football that improve over existing multi agent exploration methods.

Weaknesses:

Overall despite several reads I was not able to understand the method proposed by the authors.
The paper relies on many concepts and terms are not properly defined.
The whole architecture is never properly defined and there is also no pseudo of the algorithm developed in the paper.
Few examples:

In Definition 1, \pi_t is not defined and it is not clear that (t'', t) is an interval
In section 3.2, the mixing network, branches and local utility network are not defined.
The paper relies on CDS but it is never properly introduced, same goes for the r^short the intrinsic rewards introduced by CDS

**Summary Of The Paper:**

This paper tackles the challenge of exploration in multi agent reinforcement learning. The paper highlights the difficulty that in the multi-agent setting the value of intrinsic rewards can fluctuate and may sometimes increase, making previously visited states appear attractive again. As a solution the authors present Never Revisit an exploration technique for multi agent reinforcement learning. The authors evaluate this method on a maze environment and Google research football on show improved results over existing baselines.

**Summary Of The Review:**

Thought the paper seem to display interesting empirical results I would advise the authors to rework section 3 of their manuscript to make it easier to understand. In its current I'm unable to properly judge the paper and for that reason I cannot recommend acceptance.

---

> ### Author Response · Authors · 2022-11-15
> **Response to Reviewer XHpU**
>
> Thank you for finding our work interesting. We have updated a new version of our manuscript, especially section 3. We sincerely hope the reviewer to re-judge the value of our paper. Further feedback and discussions are appreciated.
>
> **w1: The mixing network, branches and local utility network are not defined. The paper relies on CDS but it is never properly introduced, same goes for the r^short the intrinsic rewards introduced by CDS**
>
> **A for w1:** We rework section 3 to make it easier to understand, including details of the mixing network and branches. As for CDS, we add more details about its intrinsic rewards.

---

> ### Author Response · Authors · 2022-11-17
> **Looking forward to further discussions!**
>
> Dear reviewer,
>
> We were wondering if our response and revision have resolved your concerns. In our responses and updated manuscripts, we gave a more detailed explanation of our method and we included more experiments. Please let us know if you have any further concerns, and we are looking forward to more discussions to improve our manuscript further.
>
> Best regards,
>
> The Authors

---

### Author Response · Authors · 2022-11-14
**For all reviewers, regarding clarity of the proposed method and more experiments**

We thank all reviewers for pointing out the drawbacks in presenting our method. We have submitted a new version accordingly to improve the quality of our work, including all the details of our method in a proper manner. We sincerely hope that all reviewers can read it and re-judge the value of our submission. The update could be summarized below:

* We revised the Abstract and Section 1 to make the motivation of our approach clearer.
* We supplemented a detailed explanation of labels in Figure 1 for better clarification.
* We supplemented more details of the baseline algorithm CDS, which our approach relies on, so that readers can more easily follow.
* We supplemented more details on how to estimate joint observation distributions in Section 3.1.
* We reworked Section 3.2 to thoroughly demonstrate how our approach augments neural network structures during exploration.
* We supplemented more details on how to select branches during sampling and training in Section 3.2 and Section B.1, so that readers can more easily understand our approach.
* We revised Section 3.3 to make the process of constructing sub-sampling easier to understand, which is used to adjust intrinsic motivations.

As for experiments
* We further provided visitation heat maps of our approach and CDS (in Section 4)
* We further compared our approach with baselines in the StarCraft II micromanagement challenge (SMAC) benchmark based on sparse rewards (in Section 5.2).
* To further understand the necessity of detecting revisitation, we conducted two ablations based on vanilla CDS: (1) store joint observation distributions every 100k time steps and add the KL divergence between each of them and the current one as intrinsic rewards during training, (2) estimate the distribution of all sampled joint observations and add the KL divergence between it and the current one as intrinsic rewards during training. Experiment results shown in Section 5.1 demonstrates the necessity of dynamically checking the exploration situation and making targeted adjustments when revisitation occurs.
* We further provided a 2D-Color plot of Figure 1's JS distances as shown in Figure 11 and Figure 12 (in Section C.2)
* We further demonstrated the size of replay buffer has trivial influence on revisitaion (in Section C.3)

We hope that the additional experimental results can provide more support to demonstrate the significance of our approach.

---

### Decision · Program_Chairs · 2023-01-20

**Decision:**

Reject

**Justification For Why Not Higher Score:**

The paper is unclear and even after discussion with reviewers, important questions had not been resolved. To the degree that reviewers understood the proposed method, they raised concerns about memory and computational requirements that would render the proposed approach impractical. This paper is not ready for publication.

**Justification For Why Not Lower Score:**

N/A

**Metareview: Summary, Strengths And Weaknesses:**

The paper identifies the problem of revisitation in multi-agent reinforcement learning, and aims to address this problem by extending an existing network architecture for multi-agent RL, as well as by introducing intrinsic rewards. Empirical results show that the proposed method outperforms baselines.

Reviewers for this paper found the paper well motivated, addressing an important research problem. They also considered the empirical performance achieved on Google Research Football compelling.

At the same time, all reviewers highlighted that the paper was extremely hard to follow, using undefined or vaguely defined terms and concepts and many open questions about the approach. While it is appreciated that the authors responded to the reviewer questions, the level of additional explanation needed demonstrates that the paper is not yet ready for publication. For example, the exchange with reviewer hnNg took many turns and even at the end of the exchange there were key open questions about the branching that the authors introduce as a key part of their approach.

In the discussion phase, it also became clear that the method has a very high memory and computational cost, which further limits the potential impact of the work.

All in all, the paper is not considered ready for publication at this stage.